# Learning Time-Series Representations by Hierarchical Uniformity-Tolerance Latent Balancing

**Amin Jalali\*, Milad Soltany\*, Michael Greenspan, Ali Etemad**
*{amin.jalali, milad.soltany, michael.greenspan, ali.etemad}@queensu.ca*
*Queen's University, Canada*

**Reviewed on OpenReview:** *https://openreview.net/forum?id=NTmVEAiyB5*

## Abstract

We propose TimeHUT, a novel method for learning time-series representations by hierarchical uniformity-tolerance balancing of contrastive representations. Our method uses two distinct losses to learn strong representations with the aim of striking an effective balance between uniformity and tolerance in the embedding space. First, TimeHUT uses a hierarchical setup to learn both instance-wise and temporal information from input time-series. Next, we integrate a temperature scheduler within the vanilla contrastive loss to balance the uniformity and tolerance characteristics of the embeddings. Additionally, a hierarchical angular margin loss enforces instance-wise and temporal contrast losses, creating geometric margins between positive and negative pairs of temporal sequences. This approach improves the coherence of positive pairs and their separation from the negatives, enhancing the capture of temporal dependencies within a time-series sample. We evaluate our approach on a wide range of tasks, namely 128 UCR and 30 UAE datasets for univariate and multivariate classification, as well as Yahoo and KPI datasets for anomaly detection. The results demonstrate that TimeHUT outperforms prior methods by considerable margins on classification, while obtaining competitive results for anomaly detection. Finally, detailed sensitivity and ablation studies are performed to evaluate different components and hyperparameters of our method.

## 1 Introduction

Time-series data are prevalent across diverse fields such as healthcare, meteorology, finance, smart homes, and energy applications (Yang & Hong, 2022; Hajimoradlou et al., 2022; Liu & Chen, 2024; Zheng et al., 2024). The advent of self-supervised deep learning has resulted in remarkable success in handling diverse forms of time-series due to its ability to leverage data without requiring annotations (Yang & Hong, 2022; Luo et al., 2023; Zhang et al., 2024). Contrastive methods, which enforce augmentations from the same sample to be close to each other as positive pairs while pushing augmentations from other samples apart as negative pairs, are a popular approach to self-supervised learning (Lee et al., 2024; Zerveas et al., 2021). These methods have shown strong performances in various areas including vision, language, and time-series (Dosovitskiy et al., 2020; Gao et al., 2021; Zhang et al., 2022b).

Prior works have defined two concepts that play key roles in contrastive self-supervised representation learning: *uniformity* and *tolerance* (Wang & Liu, 2021; Wang & Isola, 2020). Uniformity refers to the maximization of information and spread of representations in the latent space, while tolerance is the ability of the model to allow small variations in the input (e.g., augmentations, noise, or natural variations between semantically similar samples) without significantly altering the learned representation. While uniformity and tolerance are both desired characteristics, there exists a trade-off between the two that hinders self-supervised contrastive learning (Wang & Liu, 2021; Xiao & Lyu, 2024). An excessive focus on uniformity can result in

---

\*These authors contributed equally.

representations that are overly spread out, complicating the formation of meaningful clusters. Conversely, placing too much emphasis on tolerance can lead to an inefficient distributed representational space, with clusters potentially becoming too close or overlapping with one another. Therefore, careful consideration and balancing of these two objectives during the learning process is essential for each time-series instance and for the temporal segments within it. Accordingly, we pose the question: *How can we strike an effective balance between uniformity and tolerance to optimize the effectiveness of self-supervised contrastive learning for time-series data?*

In this paper, to address the challenge above and achieve a strong balance between uniformity and tolerance in the representation space, we propose a novel method for hierarchical instance-wise and temporal **Time**-series contrastive representation learning with **H**ierarchical **U**niformity-**T**olerance latent balancing (TimeHUT). Our method uses a hierarchical approach inspired by TS2Vec (Yue et al., 2022) and considers both instance-wise and temporal segments to learn comprehensive representations. Second, TimeHUT applies a temperature scheduling function (Kukleva et al., 2023) to the vanilla contrastive loss to balance the uniformity and tolerance characteristics of the embeddings. For a small temperature, the contrastive objective maximizes the average distance to the nearest neighbors of each sample, leading to a uniform distribution over the hypersphere. On the other hand, a large temperature maximizes the average distance over a wider range of neighbors considering more distant samples to result in tighter clusters. Our method includes a temperature scheduler that systematically explores a range of temperatures to consider different ranges of contrastive neighbors, optimizing the trade-off between achieving a uniform distribution of embeddings in the feature space and clustering of similar samples. Additionally, our method leverages a hierarchical angular margin loss to enforce instance-wise and temporal contrastive learning between positive and negative pairs of temporal sequences. The angular margin loss is inspired by face recognition literature (Boutros et al., 2022; Zhang et al., 2022a), which we adapt for time-series. This approach increases the coherence of positive pair segments and their separation from negatives within a single time-series sample to capture better temporal dependencies. We evaluate the efficacy of our model by conducting extensive experiments on univariate/multivariate time-series classification and anomaly detection, using 128 UCR and 30 UEA classification datasets along with Yahoo and KPI anomaly detection datasets. These experiments demonstrate that our method outperforms the state-of-the-art on both univariate and multivariate classification, while achieving the best or competitive results versus the state-of-the-art on anomaly detection. Detailed ablation and sensitivity studies demonstrate the impact of different components of our method.

Our contributions are summarized below:

- We present TimeHUT, a framework for learning effective time-series representations, which uses a periodic temperature scheduling function in its hierarchical contrastive loss to facilitate changing the emphasis between uniformity and tolerance of instances and their temporal segments within an instance. The temperature scheduler balances this trade-off dynamically during training, addressing the limitation of contrastive loss where a fixed temperature cannot adapt to the evolving representation space. This mechanism results in more effective learning of temporal patterns across classes while capturing distinctions between different segments within the time-series.

- TimeHUT additionally employs an instance-wise and temporal hierarchical contrastive angular margin loss. Although angular margin is well established in other areas such as face recognition, we adapt it for both temporal and instance-wise contrastive losses in time-series, enforcing better coherence among segments with close proximity, while distinctly separating them from non-neighboring ones within the same time-series sample. This adaptation enforces geometric margins between temporal segments within sequences for both temporal and instance-wise contrast losses, which remains unexplored in prior time-series work.

- Extensive experiments demonstrate that TimeHUT achieves state-of-the-art performance on classification and competitive results on anomaly detection. Our ablation study shows the combination achieves 86.4% accuracy vs. 83.0% with only hierarchical loss, demonstrating improvements beyond the sum of individual components. Our contribution lies in the novel adaptation and synergistic integration of temperature scheduling and angular margins to address the uniformity–tolerance problem in time-series representation learning. We have released our code implementation at

`https://github.com/aminjalali-research/TimeHUT` to contribute to the area and enable fast and accurate reproducibility.

## 2 Related Work

In recent years self-supervised representation learning has become an influential strategy for time-series representation learning, primarily due to its capacity to alleviate the substantial cost associated with labeling. In the following, we summarize key recent papers in this area.

**Non-contrastive methods.** A number of studies have applied non-contrastive methods such as employing encoder-decoder structures to minimize reconstruction errors for time-series representation learning (Zerveas et al., 2021; Li et al., 2023). Some have developed generative-based architectures to minimize the reconstruction error between raw data and the generated counterparts (Vaid et al., 2023). Others have utilized adversarial-based approaches with generators and discriminators for adversarial learning of time-series data (Seyfi et al., 2022; Jeon et al., 2022). Floss (Yang et al., 2023) introduces a regularizer to learn representations in the frequency domain by utilizing the periodic shift and spectral density similarity measures to learn the features with periodic consistency.

**Contrastive methods.** The fundamental concept behind contrastive learning is to maximize the similarity among different contexts of the same sample while minimizing the similarity among different samples (Luo et al., 2023; Lee et al., 2024; Nonnenmacher et al., 2022). T-Loss (Franceschi et al., 2019), for instance, uses a sub-segment of an input time-series as a positive sample with its random sub-segment while contrasting the same sub-segment with another time-series to obtain the representations. TS-TCC (Eldele et al., 2021) uses temporal contrasting to capture temporal dependencies by a cross-view prediction task. This module takes the past features of one augmentation to predict the future of another augmentation by utilizing different timestamp variations and augmentations. TS2Vec (Zerveas et al., 2021) employs instance-wise and temporal contrastive losses on two augmented sub-segments of time-series to capture both multi-resolution and multi-scale contextual details. It adopts a hierarchical approach to contrast augmented views, ensuring a robust representation for each timestamp.

TNC (Tonekaboni et al., 2021) defines temporal neighborhood boundaries in which the distribution of samples within the neighborhood vicinity is dissimilar from signals outside that neighborhood in the latent space using a de-biased contrastive loss. TimeCLR (Yang et al., 2022) obtains the invariant features through the maximization of similarity between the positive pairs and minimization of similarity between negative pairs of time-series. TF-C (Yang & Hong, 2022) focuses on capturing time-frequency consistency, where the representations derived from the time and frequency domains of a given time-series should exhibit proximity in the latent space. Another study (Hajimoradlou et al., 2022) presents a self-supervised contrastive framework that incorporates similarity distillation across both instance and temporal dimensions to pre-train the universal representations model. FEAT (Kim et al., 2023) integrates hierarchical temporal contrasting loss, feature contrasting loss, and reconstruction loss to concurrently learn feature and temporal consistency. SoftCLT (Lee et al., 2024) introduces soft assignments to sample pairs for instance-wise and temporal contrastive losses to capture the inter- and intra-temporal relationships in the data space. TimesURL (Liu & Chen, 2024) proposes frequency-temporal augmentations to preserve the temporal properties. It then constructs hard negative pairs to guide better contrastive learning along with the time reconstruction module to jointly optimize the model.

## 3 Method

### 3.1 Problem definition

Let $X = (x_1, x_2, \ldots, x_i, \ldots, x_n)$ be a time-series dataset, where $x_i \in \mathbb{R}^{T \times N}$ is a sample recorded at certain time intervals with $T$ denoting the length of the time-series, and $N$ signifying the number of variables. The full dataset $X$ contains a total number of $n$ time-series samples. Here, $N = 1$ indicates a univariate time-series dataset, while $N > 1$ indicates a multivariate time-series dataset in which inter-relationships across variables may exist. Time-series representation learning aims to develop a nonlinear embedding function $f_\theta : x \to z$,

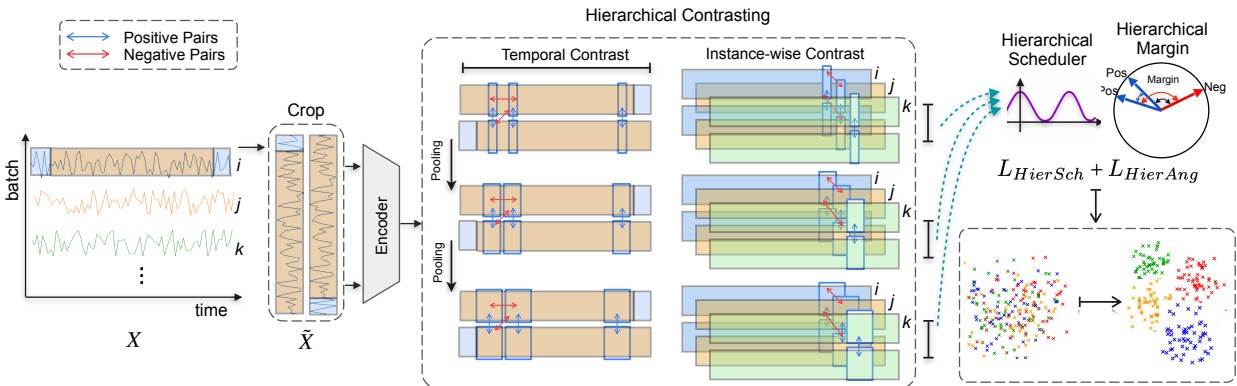

Figure 1: Overview of TimeHUT architecture. (a) Input time-series $X$ (with samples $i$, $j$, and $k$ shown in different colors) are randomly cropped into two overlapping subseries. (b) The encoder processes these subseries, followed by hierarchical contrasting at both temporal and instance-wise levels through max pooling, computing both temporal contrastive (within-sample across time) and instance-wise contrastive losses (across samples). (c) The temperature scheduler $\tau(\sigma)$ dynamically balances uniformity-tolerance trade-offs ($L_{HierSch}$), while angular margin loss enforces geometric separation between positive and negative pairs ($L_{HierAng}$). The final loss combines both components.

where $\theta$ are the learnable model parameters, and $z_i \in \mathbb{R}^{T \times M}$ are the encoded representations with $M$ denoting the dimension of the encoded features.

## 3.2 Modeling insight and theoretical foundation

Our method builds upon the hierarchical contrasting framework from TS2Vec (Yue et al., 2022), incorporating temperature scheduling inspired by Kukleva et al. (2023) and angular margin losses adapted from face recognition literature (Boutros et al., 2022; Zhang et al., 2022a). Our contribution lies in the novel adaptation and integration of these techniques for time-series representation learning to address the uniformity-tolerance problem. While TS2Vec uses fixed temperature in hierarchical contrasting, we introduce periodic scheduling (Kukleva et al., 2023) within the hierarchical framework that balances uniformity-tolerance trade-offs dynamically during training. This addresses the limitation in contrastive learning where fixed temperature parameters cannot adapt to the evolving representation space during training. The temperature parameter $\tau$ controls the boundaries between negative and positive pairs based on theoretical studies in Kukleva et al. (2023) and Wang & Isola (2020). When $\tau$ is small, the gradients are sharper and promote uniformity by maximizing average distances to the nearest neighbors. When $\tau$ is large, the gradients become smoother and promote tolerance by allowing tighter clustering.TimeHUT additionally employs instance-wise and temporal hierarchical contrastive angular margin losses to enforce coherence among proximate segments, while distinctly separating them from non-neighboring ones within the same time-series sample. This adaptation of angular margins for both temporal and instance-wise contrast losses is unexplored in prior time-series work. The combination of hierarchical temperature scheduling with the angular margins allows us to navigate the uniformity-tolerance trade-off more effectively than fixed-temperature approaches.

## 3.3 Proposed approach

The overall architecture of TimeHUT is shown in Figure 1. To learn position-agnostic representations (Kayhan & Gemert, 2020; Liu & Chen, 2024), our model processes input raw time-series $X$ by randomly cropping two overlapping subseries from the input time-series to obtain $\tilde{X}$. This is done by randomly sampling two overlapping subseries $[a_1, b_1]$ and $[a_2, b_2]$ such that $0 < a_1 \leq a_2 \leq b_1 \leq b_2 \leq T$. Subsequently, these subseries are fed into the encoder. The encoder is optimized using hierarchical contrastive loss in the temporal dimension as well as the individual instance level by summing over multiple scales (Yue et al.,

2022). The hierarchical structure captures multi-level features through maximum pooling. Next, our model applies two unique losses, which we describe in the following.

**Preliminary.** To capture the temporal characteristics of time-series, we apply a temporal contrastive loss given as:

$$L_{Temp}^{(i,t,t')} = -\log\left(\frac{\exp(s_{it,i't})}{\sum\limits_{t'\in\Omega}\left[\exp(s_{it,i't'}) + \mathbf{1}_{[t\neq t']}\exp(s_{it,it'})\right]}\right), \tag{1}$$

where $s_{it,i't} = \text{sim}(z_{i,t}, z'_{i,t})$ denotes the similarity between the representations of positive pairs at the same timestamp $t$ from the two subseries of the input time-series. Moreover, $s_{it,i't'} = \text{sim}(z_{i,t}, z'_{i,t'})$ denotes the similarity between the representations of negative pairs at different timestamps $t'$ from the two subseries and $s_{it,it'} = \text{sim}(z_{i,t}, z_{i,t'})$ represents the similarity between the representations of negative pairs at different timestamps $t'$ from the same subseries. $t$ and $i$ represent the timestamp and the index of the input time-series samples, respectively. $\text{sim}(\cdot,\cdot)$ calculates the similarity function between the embeddings of the two segments. $\Omega$ is the set of timestamps within the overlap of the two subseries, and the indicator function $\mathbf{1}_{[t\neq t']}$ is one when $t \neq t'$ and zero otherwise. In addition to $L_{Temp}$, to learn discriminate representation between different instances, the instance-wise contrastive (Inst) loss is defined by:

$$L_{Inst}^{(i,j,t)} = -\log\left(\frac{\exp(s_{it,i't})}{\sum\limits_{j=1}^{B}\left[\exp(s_{it,j't}) + \mathbf{1}_{[i\neq j]}\exp(s_{it,jt})\right]}\right), \tag{2}$$

where $B$ denotes the batch size, $s_{it,i't} = \text{sim}(z_{i,t}, z'_{i,t})$, $s_{it,j't} = \text{sim}(z_{i,t}, z'_{j,t})$, and $s_{it,jt} = \text{sim}(z_{i,t}, z_{j,t})$. $z_{i,t}$ and $z_{j,t}$ denote the representations of two different time-series at timestamp $t$ in the same batch, and $\mathbf{1}_{[i\neq j]}$ is an indicator function that is one when $i \neq j$ and zero otherwise.

This formulation for learning both instance-wise and temporal representations from time-series has been widely adopted in recent state-of-the-art solutions such as (Yue et al., 2022; Lee et al., 2024; Liu & Chen, 2024). This approach, however, does not consider the compactness and uniformity of the learned representations, and can therefore result in disrupting the semantic structure of embedding space. We address this problem by proposing the modified hierarchical learning scheme as follows.

**Uniformity-tolerance hierarchical latent balancing.** In order to allow for a gradual shift from uniformity in the embedding space to more distinct clusters, we propose a temperature parameter in the loss. A simple temperature parameter can be added to the standard $L_{Temp} + L_{Inst}$ formulation by dividing all the similarity measures $s_{it,i't}$, $s_{it,i't'}$, $s_{it,it'}$, $s_{it,j't}$, and $s_{it,jt}$ by $\tau$. We note that in this new formulation, higher temperature values will lead to tighter clusters in which the data points within a cluster are more closely packed. However, this tighter clustering comes at the cost of reduced uniformity across the entire dataset, as the data points tend to cluster more densely rather than being spread out evenly. Instead, we aim for the model to have the flexibility to move between forming well-defined clusters and ensuring there is sufficient separation between them. More specifically, we aim for the model to effectively balance the identification of discriminative and subtle instance-wise and temporal patterns through uniformity while enhancing the tolerance by bringing the features of semantically similar items close to each other (Wang & Liu, 2021; Kukleva et al., 2023; Manna et al., 2023).

To this end, we introduce a temperature scheduler $\tau(\sigma)$ to the hierarchical instance-wise and temporal contrastive losses in the context of time-series representation learning for the first time. This scheduler will enable the model to consistently shift from uniformity to tolerance without incurring extra computational expenses. Accordingly, we generate the dynamic values for the temperature scheduling mechanism by sinusoidal periodic variations through the cosine function with an amplitude and offset adjustment as $\tau(\sigma) = \Delta\tau \times \cos^2\left(\frac{\omega\sigma}{2}\right) + \tau_{min}$. The linear transformation ($\Delta\tau = \tau_{max} - \tau_{min}$) adjusts the amplitude of variations and $\tau_{min}$ shifts the entire function vertically (offset). The angular frequency is represented by $\omega = \frac{2\pi}{T}$ in which $\tau(\sigma)$ oscillates with a period $T$ and varies between $\tau_{min}$ and $\tau_{min} + \Delta\tau$. The cosine term introduces periodic oscillations with a frequency determined by period $T$. $\sigma$ represents the time variable, $\tau_{max}$ and $\tau_{min}$ are maximum and minimum values for temperature hyperparameter. The use of $\cos^2$ suggests that the parameter changes in a smooth, periodic manner, with its rate of change being slowest at the extremes and

fastest in the middle of each cycle. The time-varying temperature parameter influenced by a wave function in contrastive loss reflects how the similarity between embeddings is scaled over time, potentially adapting the learning focus from coarse to fine features. The reason for choosing the sinusoidal function is theoretically detailed in literature (Kukleva et al., 2023; Manna et al., 2023). Hence Eqs. 1 and 2 can be re-written as:

$$L_{TempSch}^{(i,t,t')} = -\log\Big(\frac{\exp(s_{it,i't}/\tau(\sigma))}{\sum\limits_{t'\in\Omega}\big[\exp(s_{it,i't'}/\tau(\sigma)) + \mathbf{1}_{[t\neq t']}\exp(s_{it,it'}/\tau(\sigma))\big]}\Big) \tag{3}$$

and

$$L_{InstSch}^{(i,j,t)} = -\log\Big(\frac{\exp(s_{it,i't})/\tau(\sigma)}{\sum\limits_{j=1}^{B}\big[\exp(s_{it,j't}/\tau(\sigma)) + \mathbf{1}_{[i\neq j]}\exp(s_{it,jt}/\tau(\sigma))\big]}\Big). \tag{4}$$

The hierarchical instance-wise and temporal loss with temperature scheduling is calculated by contrasting all samples in the batch and their temporal segments as:

$$L_{HierSch} = \frac{1}{NT}\sum_{i}\sum_{t}(L_{TempSch}^{(i,t,t')} + L_{InstSch}^{(i,j,t)}). \tag{5}$$

To further elaborate, when the temperature is small, the contrastive loss maximizes separation among embeddings to enhance uniformity. Conversely, with a large temperature, the loss encourages embeddings of similar samples to cluster tightly, promoting tolerance. By scheduling the temperature, our method explores and balances uniformity and tolerance over the training period, optimizing representational quality. The sinusoidal scheduling introduces periodic exploration of the representation space by adjusting $\tau$ in a smooth and non-monotonic manner. Furthermore, the temperature scheduler periodically modulates uniformity and tolerance. The dependence on $T$ ensures that the scheduler cycles over a temporal horizon aligned with the time-series sample. This cyclic modulation has theoretical grounding in literature (Kukleva et al., 2023; Manna et al., 2023), and empirically improves learning stability.

Given that the formulation of hierarchical contrastive learning pushes the representations of the same instances together while pushing those of different ones apart without explicitly considering semantic relationships, our model is likely to achieve higher uniformity at the cost of less tolerance. To remedy this, we take inspiration from margin separation of negative pairs (Boutros et al., 2022; Alirezazadeh & Dornaika, 2023; Terhörst et al., 2023; Zhang et al., 2022a; Choi et al., 2020), which has not been explored in this context before. Accordingly, we define a hierarchical angular margin loss to encourage the model to maintain a minimum angular distance between negative pairs of hierarchical representations, ensuring better separability in the latent space. The hierarchical angular margin loss is composed of two specific terms: the temporal angular margin loss $L_{TempAng}^{(i,t,t')}$ and the instance-wise angular margin loss $L_{InstAng}^{(i,j,t)}$. These losses hierarchically calculate the cosine similarity between segment embeddings based on the positive or negative instances to apply the marginal distance. We define $L_{TempAng}^{(i,t,t')}$ as:

$$L_{TempAng}^{(i,t,t')} = \begin{cases} (\cos^{-1}(s_{it,i't}))^2 & \text{if } \mathbb{I}(ii', tt) = 1 \\ \max(0, m_a - \cos^{-1}(s_{it,i't'}))^2 & \text{if } \mathbb{I}(ii', tt') = 1 \\ \max(0, m_a - \cos^{-1}(s_{it,it'}))^2 & \text{if } \mathbb{I}(ii, tt') = 1 \end{cases}, \tag{6}$$

where the indicator function $\mathbb{I}$ takes a condition and returns 1 if the condition is true and 0 otherwise. $\mathbb{I}(ii', tt)$ denotes an indicator for positive pair segments at the same timestamp $t$ from the two subseries $ii'$. $\mathbb{I}(ii', tt')$ denotes the negative pairs at different timestamps $t'$ from the two subseries $ii'$, highlighting those segments that are temporally distant. Moreover, $\mathbb{I}(ii, tt')$ represents the negative pairs at different timestamps $t'$ from the same subseries $ii$. The arccosine function, denoted by $\cos^{-1}$, computes the angle corresponding to the cosine similarity. The smaller the angle, the more similar the vectors are. $m_a$ is the angular margin that enforces a minimum angular separation for negative pairs.

In this formulation, if the similarity measure $s_{it,i't}$ between positive pairs increases, the loss term $L_{TempAng}^{(i,t,t')}$ will decrease as the term $\cos^{-1}(s_{it,i't})$ decreases when the embeddings become more aligned, with the angle

approaching 0 degrees. Conversely, $\cos^{-1}(s_{it,i't})$ increases as the similarity between the positive pairs deviates from perfect alignment, with the angle between the two embeddings approaching 90 degrees. This encourages the model to adjust the embeddings such that the positive pairs are pulled closer together in the feature space. Moreover, for negative pairs $\mathbb{I}(ii', tt')$ and $\mathbb{I}(ii, tt')$, when the cosine of the angle between them is less than the margin, $m_a - \cos^{-1}(s_{it,i't'}) > 0$, the model is penalized, and the gradient directs the optimization to increase the angle between them. This pushes the negative pairs further apart in the feature space, increasing $\cos^{-1}(s_{it,i't'})$ within the angular constraints imposed by $m_a$. The margin $m_a$ acts as a threshold, beyond which the loss for negative pairs does not increase. This ensures that the model enforces a minimum angular distance between the negative pairs, resulting in a structured separation in the feature space based on the temporal relationships of segments.

Next, we define $L_{InstAng}^{(i,j,t)}$ as:

$$L_{InstAng}^{(i,j,t)} = \begin{cases} (\cos^{-1}(s_{it,i't}))^2 & \text{if } \mathbb{I}(ii', tt) = 1 \\ \max(0, m_a - \cos^{-1}(s_{it,jt}))^2 & \text{if } \mathbb{I}(ij, tt) = 1 \\ \max(0, m_a - \cos^{-1}(s_{it,j't}))^2 & \text{if } \mathbb{I}(ij', tt) = 1 \end{cases} \tag{7}$$

where $\mathbb{I}(ij, tt)$ denotes two different time-series $i$ and $j$ at timestamp $t$ from the same batch. $j'$ is a cropped subseries of time-series $j$ in $\mathbb{I}(ij', tt)$ as indicated under the instance-wise contrast in Figure 1.

Here, for positive instance-wise pairs where $\mathbb{I}(ii', tt) = 1$, the loss is calculated as the square of the angle (in radians) between the embeddings. This loss function penalizes large angles between embeddings of positive pairs, encouraging the model to learn embeddings that are closely positioned in the feature space. For negative instance-wise pairs $\mathbb{I}(ij, tt)$ and $\mathbb{I}(ij', tt)$, the loss is defined as $m_a - \cos^{-1}(\text{sim}(z_{i,t}, z'_{j,t}))$ indicating that the model incurs a penalty if the angle between the embeddings is smaller than margin $m_a$. For the loss to be zero, the $m_a - \cos^{-1}(\text{sim}(z_{i,t}, z'_{j,t})) \leq 0$ condition must be satisfied. This encourages the embeddings of negative pairs to be positioned farther apart. Consequently, the loss function adjusts the embeddings such that positive pairs are separated by a small angular distance while ensuring that negative pairs are separated by an angular distance that is at least as large as the margin $m_a$.

Obtaining hierarchical instance-wise and temporal angular margin for all the time-series samples and temporal segments is defined as:

$$L_{HierAng} = \frac{1}{NT} \sum_i \sum_t (c_t L_{TempAng}^{(i,t,t')} + c_i L_{InstAng}^{(i,j,t)}), \tag{8}$$

where $c_t$ and $c_i$ are the loss term coefficients.

**Total loss.** The total loss is calculated by the sum of the hierarchical temperature scheduling loss and hierarchical angular margin loss as follows:

$$L_{Total} = L_{HierSch} + L_{HierAng}. \tag{9}$$

The intuition of our work is that the hierarchical temperature scheduler loss ($L_{HierSch}$) promotes balancing between uniformity and tolerance in the feature space enhancing the representation quality. The angular margin loss ($L_{HierAng}$) maintains distinct separations, or margins, between positive and negative pairs. This combination encourages representations that are both well-distributed and distinctive, enhancing performance. By adjusting the $L_{HierAng}$ term coefficients ($c_i$ and $c_t$), we can fine-tune the margin separation instance-wise and temporally to directly impact clustering. Angular margin loss in this context enforces a minimum angular distance between negative pairs of embeddings, setting a margin ($m_a$) to separate dissimilar instances or time points. This loss is used to improve the model's ability to handle subtle temporal dependencies and to prevent overlapping clusters.

$L_{HierSch}$ and $L_{HierAng}$ are combined with equal weighting, but each loss contains internal coefficients that can be tuned. Specifically, $L_{HierSch}$ incorporates $\tau_{min}$, $\tau_{max}$, and $T_{max}$ via its scheduler, while $L_{HierAng}$ uses coefficients $c_i$ and $c_t$ to balance instance-wise and temporal angular losses. Hence, by combining the two losses equally and relying on their internal tuning parameters, no additional hyperparameters are required.

Algorithm 1 provides PyTorch-like pseudo-code that describes the proposed TimeHUT model.

# 4   Experiment setup

**Datasets.** We evaluate TimeHUT on both univariate/multivariate classification and anomaly detection. For classification, we use the standard UCR 128 univariate dataset (Dau et al., 2019) and UEA 30 multivariate dataset (Bagnall et al., 2018). They consist of diverse time-series datasets such as ECG/EEG/MEG classification, motion classification, human activity recognition, and audio spectra classification, among others. They are widely utilized due to their ability to offer a comprehensive evaluation, measuring the model's generalization capabilities. In addition, we utilize the commonly used Yahoo (Laptev et al., 2015) and KPI (Ren et al., 2019) datasets for anomaly detection. Yahoo contains 367 hourly sampled time-series with annotated anomaly points. The KPI dataset consists of 58 univariate time-series, each representing a key performance indicator (KPI) collected from various internet companies. Each time-series is sampled at one-minute intervals. Together, all the datasets used in this paper comprise a total of 160 individual datasets.

**Implementation details.** We use Adam optimizer with a learning rate of 1e-3. The batch size is set to 8, with the number of epochs determined by the dataset size: 200 epochs for datasets smaller than 100,000, and 600 epochs for larger datasets. The representation dimension is fixed at 320. During training, we segment large time-series sequences into 3,000 timestamps, following (Yue et al., 2022; Lee et al., 2024). For the details of all the hyperparameters for all the datasets used in our study, please see Appendix C. We train our method using PyTorch 1.10 on 4 NVIDIA GeForce RTX 3090 GPUs. We used the PyHopper library (Lechner et al., 2022) to find the optimal values for each hyperparameter. It uses the Markov Chain Monte Carlo optimization algorithm to search for a range of values of $[0.2, 0.8]$ for $m_a$ and $[0, 10]$ for $c_i$ and $c_t$. It selects the values resulting in the best performance as reported in Appendix Tables A5, A6, A7, and A8.

We use a standard encoder backbone in our method following (Yue et al., 2022). This encoder consists of three components: a projection layer, a masking module, and convolution layers. The projection layer serves as a fully connected layer that transforms the input segments into high-dimensional vectors, which represent the data in a more complex space. Following this, the masking module applies masks to these vectors at randomly chosen timestamps, creating a modified version of the data. This process helps in generating an augmented view of the input time-series by selectively hiding parts of the data, encouraging the model to learn more robust features. The convolution layers incorporate 10 residual blocks. The architecture of each residual block comprises two one-dimensional convolutional layers, each defined by a dilation block that increases the perceptual field across diverse channels. The exact hyperparameters of the backbone encoder follow prior works such as (Yue et al., 2022) and (Lee et al., 2024).

**Baselines and comparisons.** To assess the performance of our proposed TimeHUT, we use accuracy and rank following the experimental setup of Pieper et al. (2023), Lee et al. (2024), and Liu & Chen (2024). For comparison, we use commonly used prior works, namely TF-C (Liu et al., 2023), MCL (Wickstrøm et al., 2022), DTW (Franceschi et al., 2019), T-Loss (Franceschi et al., 2019), TST (Zerveas et al., 2021), TS-TCC (Eldele et al., 2021), TNC (Tonekaboni et al., 2021), TS2Vec (Yue et al., 2022), InfoTS (Luo et al., 2023), InfoTS$_s$, SMDE (Zhang et al., 2024), FEAT (Kim et al., 2023), SelfDis (Pieper et al., 2023), Ti-MAE (Li et al., 2023), Floss (Yang et al., 2023), SoftCLT (Lee et al., 2024), TimesURL (Liu & Chen, 2024), and AutoTCL (Zheng et al., 2024). The InfoTS$_s$ model uses ground-truth labels only to train a meta-learner for selecting suitable augmentations, while InfoTS is entirely unsupervised. Note that among the individual datasets of UCR and UEA, some datasets cannot be handled by T-Loss, TS-TCC, TNC, and DTW due to missing observations, such as DodgerLoopDay, DodgerLoopGame, DodgerLoopWeekend, and InsectWingbeat. Therefore, we considered only 125 UCR and 29 UEA datasets to obtain the rank values following prior works (Yue et al., 2022; Lee et al., 2024; Luo et al., 2023; Zheng et al., 2024; Liu & Chen, 2024). TimeHUT works on all UCR and UEA datasets, and full results of TimeHUT on all datasets are provided in Appendix A.

In addition, following the experimental setup of TS2Vec (Yue et al., 2022), SoftCLT (Lee et al., 2024), and TimeURL (Liu & Chen, 2024) on anomaly detection, we evaluate our method using F1 score, precision, and recall metrics. We compare TimeHUT for anomaly detection with SPOT (Siffer et al., 2017), DSPOT (Siffer et al., 2017), DONUT (Xu et al., 2018), SR (Ren et al., 2019), TS2Vec (Yue et al., 2022), TimesURL (Liu & Chen, 2024), and SoftCLT (Lee et al., 2024) for the normal setting, and FFT (Rasheed et al., 2009),

Twitter-AD (Vallis et al., 2014), Luminol (LinkedIn, 2018), SR (Ren et al., 2019), TS2Vec (Yue et al., 2022), and SoftCLT (Lee et al., 2024) for the cold-start setting. We include the explanation of these methods on anomaly detection in Appendix B.

## 5  Results

We present our results on univariate and multivariate classification, as well as anomaly detection. Next, we perform ablation studies to evaluate the impact of the main components of our method. We then present a comprehensive sensitivity study for the main hyperparameters of our method.

Table 1: Performance of TimeHUT on univariate and multivariate classification.

| Model | 125 UCR Datasets | | 29 UEA Datasets | |
|---|---|---|---|---|
| | Acc | Rank | Acc | Rank |
| DTW (Franceschi et al., 2019) | 72.7 | 8.72 | 65.0 | 10.02 |
| T-Loss (Franceschi et al., 2019) | 80.6 | 6.34 | 67.5 | 8.76 |
| TST (Zerveas et al., 2021) | 64.1 | 9.79 | 63.5 | 10.97 |
| TS-TCC (Eldele et al., 2021) | 75.7 | 7.50 | 68.2 | 9.47 |
| TNC (Tonekaboni et al., 2021) | 76.1 | 7.59 | 67.7 | 10.38 |
| MCL (Wickstrøm et al., 2022) | | - | 61.2 | 12.45 |
| TS2Vec (Yue et al., 2022) | 83.0 | 4.78 | 71.2 | 7.55 |
| TF-C (Liu et al., 2023) | - | - | 42.0 | 13.53 |
| Ti-MAE (Li et al., 2023) | 82.3 | 4.77 | - | - |
| InfoTS (Luo et al., 2023) | 83.8 | 3.10 | 72.2 | 6.60 |
| InfoTS$_s$ (Luo et al., 2023) | 84.1 | 2.71 | 73.8 | 5.07 |
| FEAT (Kim et al., 2023) | - | - | 73.1 | 6.40 |
| SelfDis (Pieper et al., 2023) | 83.2 | 3.82 | 74.8 | 4.24 |
| Floss (Yang et al., 2023) | 84.9 | - | 73.9 | - |
| SoftCLT (Lee et al., 2024) | 85.0 | - | 75.1 | - |
| SMDE (Zhang et al., 2024) | - | - | 72.7 | 6.74 |
| TimesURL (Liu & Chen, 2024) | 84.5 | - | 75.2 | - |
| AutoTCL (Zheng et al., 2024) | - | - | 75.1 | 4.90 |
| TimeHUT | **86.4** | **2.02** | **77.3** | **2.93** |

**Classification.** The results of our experiments on classification are presented in Table 1, where we observe that the proposed TimeHUT model outperforms SOTA models for both univariate UCR and multivariate UEA datasets. We achieve an average accuracy and rank of 86.4% and 2.02 for UCR and 77.3% and 2.17 for UEA, respectively. The full classification results for all individual datasets within UCR and UEA are presented in Appendix A (Tables A1, A2, and A3). These results show the efficacy of hierarchical uniformity-tolerance latent balancing for classification. Note that some models are specifically designed for either classification or anomaly detection. As a result, results for both tasks were not available for all models. We also present UMAP visualizations for the learned representations by our model in Appendix E.

Table 2: TimeHUT performance compared to supervised models.

| Models | TimeHUT | HC2 | DrCIF | MultiRocket | ROCKET | HC1 |
|---|---|---|---|---|---|---|
| UEA Acc | **0.761** | 0.748 | 0.732 | - | 0.721 | 0.711 |
| UCR Acc | 0.872 | **0.876** | - | 0.868 | 0.853 | 0.865 |

Note that the models in Table 1 are all self-supervised, similar to our proposed TimeHUT. In Table 2, we extend the comparisons to fully supervised models, namely HIVE-COTE 1.0 (HC1), DrCIF, ROCKET, and HIVE-COTE 2.0 (HC2), as reported in Renault et al. (2023) and Middlehurst et al. (2021). These comparisons are conducted on 26 UEA datasets (Middlehurst et al., 2021) and 112 UCR datasets (Renault et al., 2023). The complete results for all individual datasets on 26 UEA datasets are provided in the Table 3. This experiment demonstrates that, despite our proposed TimeHUT model being trained without labels, it outperforms other models on the UEA datasets and achieves competitive performance on the UCR datasets.

Table 3: Performance comparison of TimeHUT, HC2, DrCIF, and ROCKET across UEA datasets.

| Dataset | Accuracy | | | | |
|---|---|---|---|---|---|
| | TimeHUT | HC2 | DrCIF | ROCKET | HC1 |
| ArticularyWordRecognition | 0.993 | 0.993 | 0.980 | 0.997 | 0.990 |
| AtrialFibrillation | 0.533 | 0.267 | 0.333 | 0.200 | 0.133 |
| BasicMotions | 1.000 | 1.000 | 1.000 | 1.000 | 1.000 |
| Cricket | 1.000 | 1.000 | 0.986 | 1.000 | 0.986 |
| DuckDuckGeese | 0.600 | 0.560 | 0.540 | 0.500 | 0.480 |
| ERing | 0.926 | 0.989 | 0.993 | 0.985 | 0.970 |
| EigenWorms | 0.962 | 0.947 | 0.924 | 0.908 | 0.634 |
| Epilepsy | 0.964 | 1.000 | 0.978 | 0.978 | 1.000 |
| EthanolConcentration | 0.373 | 0.772 | 0.692 | 0.445 | 0.791 |
| FaceDetection | 0.551 | 0.660 | 0.620 | 0.648 | 0.656 |
| FingerMovements | 0.642 | 0.530 | 0.600 | 0.540 | 0.550 |
| HandMovementDirection | 0.432 | 0.473 | 0.527 | 0.459 | 0.446 |
| Handwriting | 0.582 | 0.548 | 0.346 | 0.567 | 0.482 |
| Heartbeat | 0.810 | 0.732 | 0.790 | 0.741 | 0.722 |
| LSST | 0.586 | 0.643 | 0.556 | 0.622 | 0.575 |
| Libras | 0.883 | 0.933 | 0.894 | 0.900 | 0.900 |
| MotorImagery | 0.660 | 0.540 | 0.440 | 0.510 | 0.610 |
| NATOPS | 0.966 | 0.894 | 0.844 | 0.883 | 0.889 |
| PEMS-SF | 0.855 | 1.000 | 1.000 | 0.821 | 0.983 |
| PenDigits | 0.989 | 0.979 | 0.977 | 0.985 | 0.934 |
| PhonemeSpectra | 0.243 | 0.290 | 0.308 | 0.259 | 0.321 |
| RacketSports | 0.921 | 0.908 | 0.901 | 0.914 | 0.888 |
| SelfRegulationSCP1 | 0.860 | 0.891 | 0.877 | 0.843 | 0.853 |
| SelfRegulationSCP2 | 0.583 | 0.500 | 0.494 | 0.494 | 0.461 |
| StandWalkJump | 0.960 | 0.467 | 0.533 | 0.600 | 0.333 |
| UWaveGestureLibrary | 0.916 | 0.928 | 0.909 | 0.931 | 0.891 |
| Average Accuracy | 0.761 | 0.748 | 0.732 | 0.721 | 0.711 |
| Average Rank | 2.60 | 2.37 | 3.27 | 3.10 | 3.67 |

**Anomaly detection.** Table 4 presents the results for anomaly detection on Yahoo and KPI datasets. The experiments are conducted under two distinct settings following Yue et al. (2022), Lee et al. (2024), and Liu & Chen (2024). In the normal setting, each dataset is divided into two halves based on the time order, with one half used for training and the other for evaluation. In the cold-start setting, models are pre-trained on the FordA dataset from the UCR and subsequently evaluated on each individual dataset. The anomaly score is calculated as the L1 distance between two representations encoded from both masked and unmasked inputs, as explained in prior works (Yue et al., 2022; Lee et al., 2024; Liu & Chen, 2024). We observe that our proposed TimeHUT model achieves state-of-the-art performance in terms of F1 for both datasets under the normal setting, with scores of 0.755 and 0.721, respectively. Furthermore, TimeHUT attains an F1 score of 0.779 for the Yahoo dataset in the cold-start setting, demonstrating higher performance than prior works. For the KPI dataset in the cold-start setting, TimeHUT secures the second-best F1 score of 0.691.

**Forecasting.** Finally, to further evaluate our method, we present the mean squared error (MSE) of *forecasting* models including our proposed TimeHUT on the univariate ETTh$_1$ and ETTh$_2$ datasets across five prediction horizons (H = 24, 48, 168, 336, 720) in Appendix C (Table A4), which highlights the effectiveness of proposed TimeHUT.

**Ablation.** We perform ablation studies to analyze the impact of each key module in TimeHUT on 128 UCR and 30 UEA datasets, using accuracy and AUPRC metrics. First we remove the $L_{HierSch}$ and replace it with a similar loss where $\tau = 1$, effectively turning off the temperature scheduler. Next, we remove the $L_{HierSch}$ altogether. We then ablate $L_{HierAng}$ and only use the $L_{HierSch}$. And finally we remove both losses and only use $L_{HierSch}$ with $\tau = 1$. The results for this study are presented in Table 5, where we observe that each loss term used to train TimeHUT has a meaningful impact on the final outcome. Moreover, we see a positive impact for using the temperature scheduler instead of a constant temperature of $\tau = 1$.

Table 4: Performance of TimeHUT on anomaly detection.

| Model | Yahoo | | | KPI | | |
|---|---|---|---|---|---|---|
| | $F_1$ | Prec. | Rec. | $F_1$ | Prec. | Rec. |
| SPOT (Siffer et al., 2017) | 0.338 | 0.269 | 0.454 | 0.217 | 0.786 | 0.126 |
| DSPOT (Siffer et al., 2017) | 0.316 | 0.241 | 0.458 | 0.521 | 0.623 | 0.447 |
| DONUT (Xu et al., 2018) | 0.026 | 0.013 | 0.825 | 0.347 | 0.371 | 0.326 |
| SR (Ren et al., 2019) | 0.563 | 0.451 | 0.747 | 0.622 | 0.647 | 0.598 |
| TS2Vec (Yue et al., 2022) | 0.745 | 0.729 | 0.762 | 0.677 | **0.929** | 0.533 |
| SoftCLT (Lee et al., 2024) | 0.742 | 0.722 | **0.765** | 0.701 | 0.916 | 0.570 |
| TimesURL (Liu & Chen, 2024) | 0.749 | **0.748** | 0.750 | 0.688 | 0.925 | 0.546 |
| TimeHUT | **0.755** | 0.746 | 0.764 | **0.721** | 0.899 | **0.602** |
| *Cold-start*: | | | | | | |
| FFT (Rasheed et al., 2009) | 0.291 | 0.202 | 0.517 | 0.538 | 0.478 | 0.615 |
| Twitter-AD (Vallis et al., 2014) | 0.245 | 0.166 | 0.462 | 0.430 | 0.411 | 0.276 |
| Luminol (LinkedIn, 2018) | 0.388 | 0.254 | 0.818 | 0.571 | 0.478 | 0.697 |
| SR (Ren et al., 2019) | 0.529 | 0.404 | 0.765 | 0.666 | 0.670 | 0.697 |
| TS2Vec (Yue et al., 2022) | 0.726 | 0.692 | 0.763 | 0.676 | 0.907 | 0.540 |
| SoftCLT (Lee et al., 2024) | 0.762 | 0.753 | **0.773** | **0.707** | **0.921** | **0.574** |
| TimeHUT | **0.779** | **0.793** | 0.765 | 0.691 | 0.894 | 0.563 |

Table 5: Ablation experiments for the key components of TimeHUT.

| $L_{HierAng}$ | $L_{HierSch}$ | $L_{HierSch}(\tau = 1)$ | 128 UCR datasets | | 30 UEA datasets | |
|---|---|---|---|---|---|---|
| | | | Acc (%) | AUPRC | Acc (%) | AUPRC |
| ✓ | ✓ | – | **86.42** | **86.23** | **76.24** | **74.73** |
| ✓ | – | ✓ | 84.64 | 85.36 | 74.29 | 72.55 |
| ✓ | – | – | 83.36 | 83.03 | 73.71 | 73.57 |
| – | ✓ | – | 84.99 | 85.31 | 73.22 | 72.36 |
| – | – | ✓ | 83.00 | 84.21 | 71.20 | 70.74 |

**Sensitivity to hyperparameters.** We investigate the effect of the $c_t$ and $c_i$ in Equation 8 on model performance, and present four examples in Figure 2 where we plot the accuracy vs. these parameters for different datasets from UCR and UEA. We observe that while the choice of hyperparameters can expectedly have an impact on the final results, the sensitivity of our model to the optimal set of parameters is generally ⊲3%. The detailed hyperparameters for each dataset are presented in the Appendix D (Tables A5 to A8).

**Component-wise computational analysis.** We also analyze the computational cost of each component of TimeHUT model using the Chinatown dataset in Table 6. The training time overhead yields +2.03% accuracy gain, demonstrating a favorable cost-benefit trade-off.

Table 6: Component-wise computational analysis.

| Scenario | Acc | Training Time(s) | Peak GPU Memory (MB) | MFLOPs/Epoch |
|---|---|---|---|---|
| Baseline | 0.9651 | 9.36 | 1330.6 | 51.2 |
| +HierAng$_{Instance}$ | 0.9823 | 11.09 | 1315.9 | 60.5 |
| +HierAng$_{Temporal}$ | 0.9795 | 12.34 | 1341.6 | 62.7 |
| +HierAng$_{Both}$ | 0.9828 | 12.49 | 1367.6 | 65.3 |
| +HierSch | 0.9815 | 9.53 | 1332.2 | 58.5 |
| +HierAng$_{Both}$ + HierSch | **0.9854** | 12.95 | 1398.4 | 69.1 |

**Comparison with different temperature schedulers.** We compare various temperature scheduling strategies on the Chinatown and AtrialFibrillation datasets. The base hyperparameters for all schedulers are $\tau_{min} = 0.1$, $\tau_{max} = 0.75$, and $T_{max} = 10$. As shown in Table 7, our Hierarchical-Scheduler achieves the best accuracy-efficiency trade-off.

Table 7: Comparison of different temperature schedulers on UCR Chinatown and UEA AtrialFibrillation datasets reporting accuracy, training time, and peak GPU memory values.

| Scheduler | HyperParams | Chinatown | | | AtrialFibrillation | | |
|---|---|---|---|---|---|---|---|
| | | Acc | Time(s) | GPU(MB) | Acc | Time(s) | GPU(MB) |
| Exponential | Decay=0.95 | 0.980 | 12.83 | 1376.8 | 0.417 | 20.32 | 3081.4 |
| Sigmoid | Steep=1 | 0.979 | 14.11 | 1421.5 | 0.412 | 20.31 | 3901.2 |
| Warmup-Cosine | Warmup=2 | 0.962 | 12.43 | 1414.9 | 0.407 | 19.76 | 2755.3 |
| Sawtooth-Cyclic | Cycle=T/3 | 0.979 | 12.48 | 1392.1 | 0.413 | 19.93 | 3551.6 |
| Logarithmic | Offset=1 | 0.980 | 12.85 | 1366.2 | 0.425 | 20.18 | 3405.1 |
| Step-Decay | Gamma=0.5 | 0.977 | 12.41 | 1406.3 | 0.468 | 19.89 | 3385.9 |
| Cosine-Restarts | Period=5 | 0.974 | 12.58 | 1387.7 | 0.446 | 20.57 | 3249.2 |
| Hyperbolic-Tangent | Steep=2 | 0.973 | 12.43 | 1393.8 | 0.467 | 20.72 | 3251.1 |
| **Hierarchical-Scheduler** | – | **0.985** | 12.95 | 1398.4 | **0.534** | 20.13 | 3171.7 |

**Computational efficiency analysis.** We provide an efficiency analysis comparing TimeHUT with other baselines on the UCR Chinatown dataset using NVIDIA RTX 3090. As shown in Table 8, TimeHUT's training time (12.95s) compared to TS2Vec (9.36s) is justified by accuracy improvements from 0.965 to 0.985. On this particular dataset, TS-TCC shows comparable results to TimeHUT; however, according to Table 1, TS-TCC's average accuracies on 125 UCR and 29 UEA are (75.7, 68.2), which are lower than TimeHUT (86.4, 77.3).

Table 8: Efficiency analysis of TimeHUT vs. other baselines.

| Model | Acc | F1-Score | GPU Mem(MB) | MFLOPs/Epoch | Training Time(s) |
|---|---|---|---|---|---|
| TF-C (Liu et al., 2023) | 0.8663 | 0.8514 | 1395.4 | 72.4 | 10.15 |
| MF-CLR (Duan et al., 2024) | 0.8861 | 0.8726 | 1376.6 | 52.3 | 9.55 |
| CPC (Oord et al., 2018) | 0.9301 | 0.9181 | 1377.4 | 120.2 | 14.2 |
| T-Loss (Franceschi et al., 2019) | 0.9511 | 0.8191 | 1594.4 | 56.3 | 164.3 |
| TS2Vec (Yue et al., 2022) | 0.9651 | 0.9684 | 1330.6 | 51.2 | 9.36 |
| CoST (Woo et al., 2022) | 0.9704 | 0.9609 | 1420.6 | 240.6 | 19.58 |
| TNC (Tonekaboni et al., 2021) | 0.9772 | 0.9472 | 1414.1 | 192.1 | 35.91 |
| TimesURL (Liu & Chen, 2024) | 0.9744 | 0.9756 | 1392.7 | 113.4 | 22.62 |
| TS-TCC (Eldele et al., 2021) | 0.9831 | 0.9784 | 1399.4 | 70.4 | 13.14 |
| TimeHUT | **0.9854** | **0.9826** | 1398.4 | 69.1 | 12.95 |

**Wilcoxon rank-sum statistical test.** Following Ismail-Fawaz et al. (2023), we obtain the difference in average accuracies of the 30 UEA datasets for each pair of models. We then count how often Model A achieves higher accuracy than Model B (wins), how often they are equal (draws), and how often Model B achieves higher accuracy (losses). Moreover, we calculate the statistical significance of the difference between the two models by performing the Wilcoxon signed-rank test across the 30 datasets. The p-values from the Wilcoxon signed-rank test are calculated to assess whether there is a statistically significant difference between the median of a paired dataset (e.g., $p < 0.05$). We show the Mean-Difference (MD), Wins/Draw/Losses (W/D/L), and Wilcoxon p-value in Table 9. The complete results for all individual models are provided in the Appendix (Tables A9 and A10).

**Uniformity vs. tolerance trade-off analysis.** Figure 3 presents the comparison of uniformity and tolerance across multiple datasets (e.g., Chinatown, Computers, Dog5LoopWeekend, and HouseTwenty) for three different methods: Random initialization, TS2Vec, and TimeHUT. We observe a trade-off between uniformity and tolerance where methods with higher uniformity values (e.g., TS2Vec) tend to exhibit lower tolerance values. This occurs because spreading embeddings in the latent space can inadvertently separate similar data points. In contrast, Random initialization shows higher tolerance but lower uniformity, as preserving local relationships may result in clustering within the embedding space. This trade-off highlights the need for a balance between uniformity and tolerance to produce embeddings that generalize effectively. Notably, our TimeHUT model archives a balanced trade-off between uniformity and tolerance across most datasets.

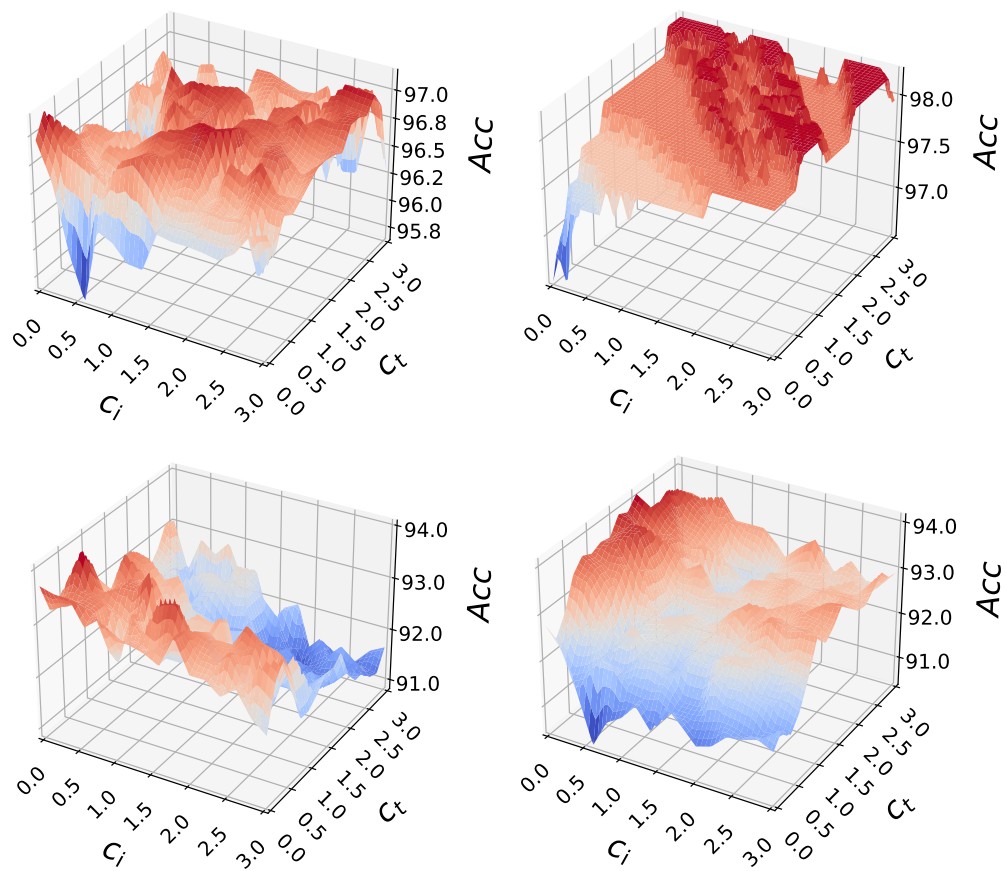

Figure 2: The accuracy of TimeHUT versus different values of $c_i$ and $c_t$, on the "Star Light Curves" dataset (top left), "China Town" dataset (top right), "Ford-A" dataset (bottom left), and "None-Invasive Fetal ECG Thorax 1" dataset (bottom right).

Table 9: Performance comparison of various metrics for TimeHUT, including Mean Difference (MD), Wins/Draws/Losses (W/D/L), and statistical significance (p-value).

| TimeHUT vs. | MD | W/D/L | p-value |
|---|---|---|---|
| AutoTCL (Zheng et al., 2024) | 0.02 | 20/4/6 | 0.0188 |
| SelfDis (Pieper et al., 2023) | 0.02 | 18/3/9 | 0.1073 |
| InfoTS$_s$ (Luo et al., 2023) | 0.03 | 17/5/8 | 0.0091 |
| FEAT (Kim et al., 2023) | 0.04 | 25/2/3 | <1e-4 |
| SMDE (Zhang et al., 2024) | 0.04 | 23/2/5 | 0.001 |
| InfoTS (Luo et al., 2023) | 0.05 | 21/2/7 | 0.0002 |
| TS2Vec (Yue et al., 2022) | 0.06 | 25/4/1 | <1e-4 |
| TNC (Tonekaboni et al., 2021) | 0.09 | 28/0/2 | <1e-4 |
| TS-TCC (Eldele et al., 2021) | 0.09 | 28/1/1 | <1e-4 |
| T-Loss (Franceschi et al., 2019) | 0.10 | 25/2/3 | <1e-4 |
| TST (Zerveas et al., 2021) | 0.15 | 28/1/1 | <1e-4 |
| MCL (Wickstrøm et al., 2022) | 0.16 | 30/0/0 | <1e-4 |
| TF-C (Liu et al., 2023) | 0.35 | 29/0/1 | <1e-4 |

**Failure cases.** We identify scenarios where TimeHUT encounters limitations. For short sequences ($T < 20$), the hierarchical contrasting has insufficient temporal context, resulting in suboptimal performance. In the case of very long sequences ($T > 10,000$), the quadratic complexity with respect to sequence length becomes computationally prohibitive. On the other hand, when only a limited number of samples per class are

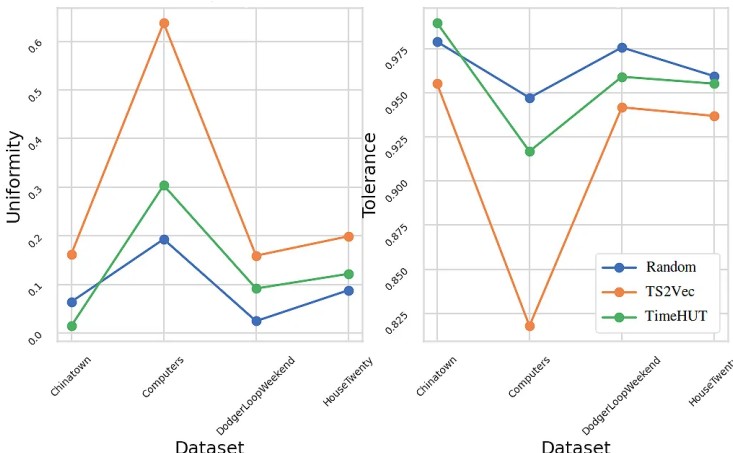

Figure 3: Uniformity and tolerance values for different datasets.

available, the angular margins may become overly restrictive, potentially overseparating natural clusters. In our experiments, TimeHUT showed reduced improvements on datasets such as ScreenType (accuracy: 0.403) and PhonemeSpectra (accuracy: 0.243), which have either very short sequences or complex multivariate distributions where angular margins may overseparate natural clusters.

## 6  Conclusions

In this work, we propose TimeHUT to learn effective time-series representations. Our proposed method uses two hierarchical loss terms to strike a balance between uniformity and tolerance in the embedding space with the goal of maximizing performance. Our method's use of hierarchical instance-wise and temporal angular margin loss and hierarchical temperature scheduling effectively captures temporal dependencies and distinguishes between positive and negative sequence pairs. The extensive experimentation conducted across hundreds of datasets demonstrated the strong performance of TimeHUT w.r.t. existing methods. Sensitivity and ablation studies are conducted to assess the impact of different components and hyperparameters.

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

**Appendix**

## A    Additional results

Tables A1 and A2 present the full classification results of our method for all the individual datasets on 128 UCR datasets, compared to related prior works, including TST (Zerveas et al., 2021), DTW (Franceschi et al., 2019), TS-TCC (Eldele et al., 2021), TNC (Tonekaboni et al., 2021), T-Loss (Franceschi et al., 2019), Ti-MAE (Li et al., 2023), TS2Vec (Yue et al., 2022), SelfDis (Pieper et al., 2023), FEAT (Kim et al., 2023), and InfoTS (Luo et al., 2023). Among these baselines, TimeHUT achieves the best accuracy on average. Besides, the full results of TimeHUT for 30 multivariate datasets in the UEA archive are also provided in Table A3 including models such as AutoTCL (Zheng et al., 2024), SMDE (Zhang et al., 2024), TF-C (Liu et al., 2023), and MCL (Wickstrøm et al., 2022), where TimeHUT provides the best average accuracy and average rank. Note that these tables do not contain certain works such as Floss (Yang et al., 2023), SoftCLT (Lee et al., 2024), and TimesURL (Liu & Chen, 2024) given that the breakdown results for all the datasets are not available.

## B    Related works on anomaly detection

A variety of methods have been developed to address anomaly detection in time series data, each offering unique approaches to identifying and analyzing outliers in streaming and static datasets. SPOT (Siffer et al., 2017) is an outlier detection method for streaming univariate time series. It leverages Extreme Value Theory and is not based on preset thresholds or predefined data distributions. It requires only a single parameter to manage the number of false positives. DONUT (Xu et al., 2018) is an unsupervised anomaly detection algorithm that uses a variational autoencoder. SR (Ren et al., 2019) is designed for time-series anomaly detection, utilizing the spectral residual model in combination with a Convolutional Neural Network. The SR model, originally used in visual saliency detection, enhances the algorithm's performance when paired with CNN. FFT (Rasheed et al., 2009) utilizes the fast Fourier transform to identify regions with high-frequency changes. Twitter-AD (Vallis et al., 2014) automatically detects long-term anomalies by recognizing the irregularities in application and system metrics of cloud data. Luminol (LinkedIn, 2018) is a Python library for time series data analysis, offering two primary features: anomaly detection and anomaly correlation to analyze the potential causes of anomalies. TS2Vec (Zerveas et al., 2021) employs temporal- and instance-wise contrastive losses on two augmented subseries of time-series to capture multi-scale contextual details for anomaly detection. SoftCLT (Lee et al., 2024) introduces soft assignments to sample pairs for hierarchical contrastive losses to capture the inter- and intra-temporal relationships in the data space. TimesURL (Liu & Chen, 2024) proposes frequency-temporal augmentations to preserve the temporal properties. It constructs hard negative pairs to guide better contrastive learning along with the time reconstruction module to jointly optimize the model for anomaly detection.

## C    Forecasting.

Table A4 presents the mean squared error (MSE) of forecasting models including our proposed TimeHUT on the univariate $ETTh_1$ and $ETTh_2$ datasets across five prediction horizons (H = 24, 48, 168, 336, 720). TimeHUT consistently achieves the lowest MSE at every horizon and on both datasets (e.g., 0.037 vs. 0.039 for H = 24 on $ETTh_1$, and 0.090 vs. 0.090 for H = 24 on $ETTh_2$), with the performance at longer horizons (e.g., 0.115 vs. 0.154 at H = 336 on $ETTh_1$ and 0.195 vs. 0.213 on $ETTh_2$), demonstrating its ability to capture both short- and long-term dependencies. Averaged over all ten configurations, TimeHUT attains an MSE of 0.125, an 8% reduction relative to the TS2Vec method, which highlights the effectiveness of our proposed TimeHUT method across multiple scales.

## D    Hyperparameters

We present the values of hyperparameters including $c_i$, $c_t$, $m_a$, $\tau_{min}$, $\tau_{max}$, and $T_{max}$ for the TimeHUT model across different datasets. Specifically, these values are detailed in Tables A5 and A6 for 128 UCR datasets, in Table A7 for 30 UEA datasets, and in Table A8 for anomaly detection.

## E    UMAP visualization

In Figure A1, we present UMAP visualizations for the learned time-series representations by our model. The top-left and bottom-left figures display the representations generated by TS2Vec for the UEA "Basic Motion" dataset and the UCR "Dodger Loop Weekend" dataset, respectively. On the right, we show the representations learned by TimeHUT. Each color corresponds to a different class. We observe that the learned representations on the right are more distinctly separated and form tighter clusters.

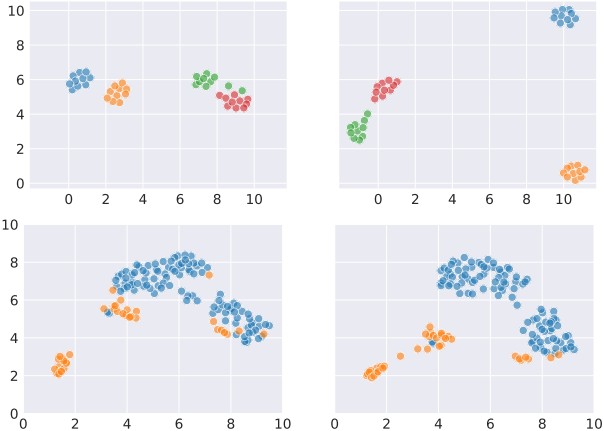

Figure A1: UMAP visualization of the learned representations by TimeHUT (right) vs. TS2Vec (left) on "Basic Motion" (top row) and "Dodger Loop Weekend" (bottom row) datasets.

Table A1: Performance of TimeHUT on the 128 individual datasets from UCR (Part 1).

| Dataset | TimeHUT | InfoTS$_s$ | InfoTS | SelfDis | TS2Vec | Ti-MAE | T-Loss | TNC | TS-TCC | DTW | TST |
|---|---|---|---|---|---|---|---|---|---|---|---|
| ACSF1 | 0.920 | 0.850 | 0.850 | 0.830 | 0.900 | 0.820 | 0.900 | 0.730 | 0.730 | 0.640 | 0.760 |
| Adiac | 0.783 | 0.795 | 0.788 | 0.788 | 0.762 | 0.788 | 0.675 | 0.726 | 0.767 | 0.604 | 0.550 |
| AllGestureWiimoteX | 0.796 | 0.560 | 0.630 | 0.776 | 0.777 | 0.633 | 0.763 | 0.703 | 0.697 | 0.716 | 0.259 |
| AllGestureWiimoteY | 0.813 | 0.623 | 0.686 | 0.777 | 0.793 | 0.682 | 0.726 | 0.699 | 0.741 | 0.729 | 0.423 |
| AllGestureWiimoteZ | 0.793 | 0.633 | 0.629 | 0.749 | 0.746 | 0.671 | 0.723 | 0.646 | 0.689 | 0.643 | 0.447 |
| ArrowHead | 0.863 | 0.874 | 0.874 | 0.857 | 0.857 | 0.874 | 0.766 | 0.703 | 0.737 | 0.703 | 0.771 |
| BME | 1.000 | 1.000 | 1.000 | 1.000 | 0.993 | 1.000 | 0.993 | 0.973 | 0.933 | 0.900 | 0.760 |
| Beef | 0.833 | 0.900 | 0.833 | 0.667 | 0.767 | 0.900 | 0.667 | 0.733 | 0.600 | 0.633 | 0.500 |
| BeetleFly | 0.900 | 0.950 | 0.950 | 0.950 | 0.900 | 0.900 | 0.800 | 0.850 | 0.800 | 0.700 | 1.000 |
| BirdChicken | 1.000 | 0.850 | 0.900 | 0.800 | 0.800 | 1.000 | 0.850 | 0.750 | 0.650 | 0.750 | 0.650 |
| CBF | 1.000 | 1.000 | 0.999 | 0.997 | 1.000 | 1.000 | 0.983 | 0.983 | 0.998 | 0.997 | 0.898 |
| Car | 0.933 | 0.900 | 0.883 | 0.850 | 0.833 | 0.867 | 0.833 | 0.683 | 0.583 | 0.733 | 0.550 |
| Chinatown | 0.985 | 0.985 | 0.988 | 0.965 | 0.965 | 0.985 | 0.951 | 0.977 | 0.983 | 0.957 | 0.936 |
| ChlorineConcentration | 0.842 | 0.825 | 0.822 | 0.754 | 0.832 | 0.725 | 0.749 | 0.760 | 0.753 | 0.648 | 0.562 |
| CinCECGTorso | 0.848 | 0.896 | 0.928 | 0.654 | 0.827 | 0.971 | 0.713 | 0.669 | 0.671 | 0.651 | 0.508 |
| Coffee | 1.000 | 1.000 | 1.000 | 1.000 | 1.000 | 1.000 | 1.000 | 1.000 | 1.000 | 1.000 | 0.821 |
| Computers | 0.716 | 0.720 | 0.748 | 0.772 | 0.660 | 0.780 | 0.664 | 0.684 | 0.704 | 0.700 | 0.696 |
| CricketX | 0.821 | 0.780 | 0.774 | 0.767 | 0.782 | 0.674 | 0.713 | 0.623 | 0.731 | 0.754 | 0.385 |
| CricketY | 0.800 | 0.774 | 0.774 | 0.751 | 0.749 | 0.659 | 0.728 | 0.597 | 0.718 | 0.744 | 0.467 |
| CricketZ | 0.818 | 0.792 | 0.787 | 0.754 | 0.792 | 0.718 | 0.708 | 0.682 | 0.713 | 0.754 | 0.403 |
| Crop | 0.768 | 0.766 | 0.766 | 0.763 | 0.756 | 0.751 | 0.722 | 0.738 | 0.742 | 0.665 | 0.710 |
| DiatomSizeReduction | 0.990 | 0.997 | 0.997 | 0.987 | 0.984 | 0.984 | 0.984 | 0.993 | 0.977 | 0.967 | 0.961 |
| DistalPhalanxOutlineAgeGroup | 0.734 | 0.763 | 0.763 | 0.755 | 0.727 | 0.763 | 0.727 | 0.741 | 0.755 | 0.770 | 0.741 |
| DistalPhalanxOutlineCorrect | 0.793 | 0.808 | 0.801 | 0.804 | 0.761 | 0.793 | 0.775 | 0.754 | 0.754 | 0.717 | 0.728 |
| DistalPhalanxTW | 0.734 | 0.720 | 0.727 | 0.748 | 0.698 | 0.727 | 0.676 | 0.669 | 0.676 | 0.590 | 0.568 |
| ECG200 | 0.930 | 0.950 | 0.930 | 0.940 | 0.920 | 0.910 | 0.940 | 0.830 | 0.880 | 0.770 | 0.830 |
| ECG5000 | 0.945 | 0.945 | 0.945 | 0.940 | 0.935 | 0.942 | 0.933 | 0.937 | 0.941 | 0.924 | 0.928 |
| ECGFiveDays | 1.000 | 1.000 | 1.000 | 1.000 | 1.000 | 0.988 | 1.000 | 0.999 | 0.878 | 0.768 | 0.763 |
| EOGHorizontalSignal | 0.594 | 0.577 | 0.572 | 0.608 | 0.539 | 0.558 | 0.605 | 0.442 | 0.401 | 0.503 | 0.373 |
| EOGVerticalSignal | 0.533 | 0.459 | 0.459 | 0.475 | 0.503 | 0.547 | 0.434 | 0.392 | 0.376 | 0.448 | 0.298 |
| Earthquakes | 0.827 | 0.821 | 0.821 | 0.748 | 0.748 | 0.748 | 0.748 | 0.748 | 0.748 | 0.719 | 0.748 |
| ElectricDevices | 0.745 | 0.691 | 0.702 | 0.704 | 0.721 | 0.685 | 0.707 | 0.700 | 0.686 | 0.602 | 0.676 |
| EthanolLevel | 0.588 | 0.710 | 0.712 | 0.670 | 0.468 | 0.744 | 0.382 | 0.424 | 0.486 | 0.276 | 0.260 |
| FaceAll | 0.914 | 0.929 | 0.929 | 0.873 | 0.771 | 0.880 | 0.786 | 0.766 | 0.813 | 0.808 | 0.504 |
| FaceFour | 0.955 | 0.864 | 0.818 | 0.795 | 0.932 | 0.875 | 0.920 | 0.659 | 0.773 | 0.830 | 0.511 |
| FacesUCR | 0.945 | 0.917 | 0.913 | 0.908 | 0.924 | 0.866 | 0.884 | 0.789 | 0.863 | 0.905 | 0.543 |
| FiftyWords | 0.802 | 0.809 | 0.793 | 0.769 | 0.771 | 0.787 | 0.732 | 0.653 | 0.653 | 0.690 | 0.525 |
| Fish | 0.966 | 0.949 | 0.937 | 0.937 | 0.926 | 0.897 | 0.891 | 0.817 | 0.817 | 0.920 | 0.720 |
| FordA | 0.936 | 0.925 | 0.915 | 0.930 | 0.936 | 0.818 | 0.928 | 0.902 | 0.930 | 0.555 | 0.568 |
| FordB | 0.817 | 0.795 | 0.785 | 0.790 | 0.794 | 0.652 | 0.793 | 0.733 | 0.815 | 0.620 | 0.507 |
| FreezerRegularTrain | 0.994 | 0.998 | 0.996 | 0.998 | 0.986 | 0.987 | 0.956 | 0.991 | 0.989 | 0.899 | 0.922 |
| FreezerSmallTrain | 0.988 | 0.991 | 0.988 | 0.980 | 0.870 | 0.959 | 0.933 | 0.982 | 0.979 | 0.753 | 0.920 |
| Fungi | 1.000 | 0.866 | 0.946 | 0.989 | 0.957 | 0.968 | 1.000 | 0.527 | 0.753 | 0.839 | 0.366 |
| GestureMidAirD1 | 0.700 | 0.592 | 0.592 | 0.654 | 0.608 | 0.662 | 0.608 | 0.431 | 0.369 | 0.569 | 0.208 |
| GestureMidAirD2 | 0.608 | 0.459 | 0.492 | 0.631 | 0.469 | 0.546 | 0.546 | 0.362 | 0.254 | 0.608 | 0.138 |
| GestureMidAirD3 | 0.438 | 0.323 | 0.315 | 0.331 | 0.292 | 0.400 | 0.285 | 0.292 | 0.177 | 0.323 | 0.154 |
| GesturePebbleZ1 | 0.913 | 0.895 | 0.802 | 0.738 | 0.930 | 0.901 | 0.919 | 0.378 | 0.395 | 0.791 | 0.500 |
| GesturePebbleZ2 | 0.943 | 0.905 | 0.842 | 0.677 | 0.873 | 0.918 | 0.899 | 0.316 | 0.430 | 0.671 | 0.380 |
| GunPoint | 0.993 | 1.000 | 1.000 | 1.000 | 0.980 | 0.993 | 0.980 | 0.967 | 0.993 | 0.907 | 0.827 |
| GunPointAgeSpan | 0.997 | 0.997 | 1.000 | 0.994 | 0.987 | 0.994 | 0.994 | 0.984 | 0.994 | 0.918 | 0.991 |
| GunPointMaleVersusFemale | 1.000 | 1.000 | 1.000 | 1.000 | 1.000 | 0.997 | 0.997 | 0.994 | 0.997 | 0.997 | 1.000 |
| GunPointOldVersusYoung | 1.000 | 1.000 | 1.000 | 1.000 | 1.000 | 1.000 | 1.000 | 1.000 | 1.000 | 0.838 | 1.000 |
| Ham | 0.781 | 0.848 | 0.838 | 0.733 | 0.714 | 0.800 | 0.724 | 0.752 | 0.743 | 0.467 | 0.524 |
| HandOutlines | 0.946 | 0.946 | 0.946 | 0.916 | 0.922 | 0.919 | 0.922 | 0.930 | 0.724 | 0.881 | 0.735 |
| Haptics | 0.549 | 0.545 | 0.546 | 0.464 | 0.526 | 0.484 | 0.490 | 0.474 | 0.396 | 0.377 | 0.357 |
| Herring | 0.703 | 0.703 | 0.656 | 0.641 | 0.641 | 0.656 | 0.594 | 0.594 | 0.594 | 0.531 | 0.594 |
| HouseTwenty | 0.983 | 0.941 | 0.924 | 0.941 | 0.916 | 0.941 | 0.933 | 0.782 | 0.790 | 0.924 | 0.815 |
| InlineSkate | 0.456 | 0.420 | 0.424 | 0.471 | 0.415 | 0.380 | 0.371 | 0.378 | 0.347 | 0.384 | 0.287 |
| InsectEPGRegularTrain | 1.000 | 1.000 | 1.000 | 1.000 | 1.000 | 1.000 | 1.000 | 1.000 | 1.000 | 0.872 | 1.000 |
| InsectEPGSmallTrain | 1.000 | 1.000 | 1.000 | 1.000 | 1.000 | 1.000 | 1.000 | 1.000 | 1.000 | 0.735 | 1.000 |
| InsectWingbeatSound | 0.644 | 0.664 | 0.639 | 0.590 | 0.630 | 0.639 | 0.597 | 0.549 | 0.415 | 0.355 | 0.266 |
| ItalyPowerDemand | 0.971 | 0.971 | 0.966 | 0.963 | 0.925 | 0.967 | 0.954 | 0.928 | 0.955 | 0.950 | 0.845 |
| LargeKitchenAppliances | 0.885 | 0.851 | 0.853 | 0.861 | 0.845 | 0.787 | 0.789 | 0.776 | 0.848 | 0.795 | 0.595 |
| Lightning2 | 0.934 | 0.934 | 0.934 | 0.902 | 0.869 | 0.836 | 0.869 | 0.869 | 0.836 | 0.869 | 0.705 |
| Lightning7 | 0.877 | 0.863 | 0.877 | 0.808 | 0.863 | 0.808 | 0.795 | 0.767 | 0.685 | 0.726 | 0.411 |
| Mallat | 0.969 | 0.967 | 0.974 | 0.950 | 0.914 | 0.956 | 0.951 | 0.871 | 0.922 | 0.934 | 0.713 |

Table A2: Performance of TimeHUT on the 128 individual datasets from UCR (Part 2).

| Dataset | TimeHUT | InfoTS$_s$ | InfoTS | SelfDis | TS2Vec | Ti-MAE | T-Loss | TNC | TS-TCC | DTW | TST |
|---|---|---|---|---|---|---|---|---|---|---|---|
| Meat | 0.967 | 0.967 | 0.967 | 0.967 | 0.950 | 0.967 | 0.950 | 0.917 | 0.883 | 0.933 | 0.900 |
| MedicalImages | 0.828 | 0.920 | 0.820 | 0.803 | 0.789 | 0.771 | 0.750 | 0.754 | 0.747 | 0.737 | 0.632 |
| MelbournePedestrian | 0.964 | 0.964 | 0.962 | 0.958 | 0.959 | 0.949 | 0.944 | 0.942 | 0.949 | 0.791 | 0.741 |
| MiddlePhalanxOutlineAgeGroup | 0.662 | 0.662 | 0.662 | 0.649 | 0.636 | 0.675 | 0.656 | 0.643 | 0.630 | 0.500 | 0.617 |
| MiddlePhalanxOutlineCorrect | 0.869 | 0.859 | 0.859 | 0.852 | 0.838 | 0.811 | 0.825 | 0.818 | 0.818 | 0.698 | 0.753 |
| MiddlePhalanxTW | 0.617 | 0.636 | 0.617 | 0.623 | 0.584 | 0.623 | 0.591 | 0.571 | 0.610 | 0.506 | 0.506 |
| MixedShapesRegularTrain | 0.929 | 0.940 | 0.935 | 0.922 | 0.917 | 0.922 | 0.905 | 0.911 | 0.855 | 0.842 | 0.879 |
| MixedShapesSmallTrain | 0.894 | 0.892 | 0.887 | 0.877 | 0.861 | 0.875 | 0.860 | 0.813 | 0.735 | 0.780 | 0.828 |
| MoteStrain | 0.925 | 0.873 | 0.873 | 0.880 | 0.861 | 0.913 | 0.851 | 0.825 | 0.843 | 0.835 | 0.768 |
| NonInvasiveFetalECGThorax1 | 0.948 | 0.941 | 0.941 | 0.924 | 0.930 | 0.918 | 0.878 | 0.898 | 0.898 | 0.790 | 0.471 |
| NonInvasiveFetalECGThorax2 | 0.953 | 0.943 | 0.944 | 0.930 | 0.938 | 0.938 | 0.919 | 0.912 | 0.913 | 0.865 | 0.832 |
| OSULeaf | 0.864 | 0.760 | 0.760 | 0.806 | 0.851 | 0.736 | 0.760 | 0.723 | 0.723 | 0.591 | 0.545 |
| OliveOil | 0.933 | 0.933 | 0.933 | 0.867 | 0.900 | 0.933 | 0.867 | 0.833 | 0.800 | 0.833 | 0.800 |
| PLAID | 0.553 | 0.356 | 0.355 | 0.449 | 0.561 | 0.458 | 0.555 | 0.495 | 0.445 | 0.840 | 0.419 |
| PhalangesOutlinesCorrect | 0.819 | 0.826 | 0.826 | 0.834 | 0.809 | 0.772 | 0.784 | 0.787 | 0.804 | 0.728 | 0.773 |
| Phoneme | 0.320 | 0.272 | 0.281 | 0.266 | 0.312 | 0.229 | 0.276 | 0.180 | 0.242 | 0.228 | 0.139 |
| PickupGestureWiimoteZ | 0.940 | 0.820 | 0.820 | 0.700 | 0.820 | 0.840 | 0.740 | 0.620 | 0.600 | 0.660 | 0.240 |
| PigAirwayPressure | 0.817 | 0.433 | 0.432 | 0.793 | 0.630 | 0.240 | 0.510 | 0.413 | 0.380 | 0.106 | 0.120 |
| PigArtPressure | 0.971 | 0.820 | 0.830 | 0.904 | 0.966 | 0.760 | 0.928 | 0.808 | 0.524 | 0.245 | 0.774 |
| PigCVP | 0.928 | 0.654 | 0.653 | 0.889 | 0.812 | 0.750 | 0.788 | 0.649 | 0.615 | 0.154 | 0.596 |
| Plane | 1.000 | 1.000 | 1.000 | 1.000 | 1.000 | 1.000 | 0.990 | 1.000 | 1.000 | 1.000 | 0.933 |
| PowerCons | 1.000 | 0.995 | 1.000 | 0.961 | 0.961 | 1.000 | 0.900 | 0.933 | 0.961 | 0.878 | 0.911 |
| ProximalPhalanxOutlineAgeGroup | 0.859 | 0.883 | 0.883 | 0.883 | 0.834 | 0.863 | 0.844 | 0.854 | 0.839 | 0.805 | 0.854 |
| ProximalPhalanxOutlineCorrect | 0.914 | 0.924 | 0.927 | 0.883 | 0.887 | 0.876 | 0.859 | 0.866 | 0.873 | 0.784 | 0.770 |
| ProximalPhalanxTW | 0.849 | 0.849 | 0.844 | 0.824 | 0.824 | 0.829 | 0.771 | 0.810 | 0.800 | 0.761 | 0.780 |
| RefrigerationDevices | 0.645 | 0.624 | 0.624 | 0.571 | 0.589 | 0.611 | 0.515 | 0.565 | 0.563 | 0.464 | 0.483 |
| Rock | 0.820 | 0.760 | 0.760 | 0.840 | 0.700 | 0.660 | 0.580 | 0.580 | 0.600 | 0.600 | 0.680 |
| ScreenType | 0.403 | 0.510 | 0.493 | 0.480 | 0.411 | 0.579 | 0.416 | 0.509 | 0.419 | 0.397 | 0.419 |
| SemgHandGenderCh2 | 0.973 | 0.939 | 0.944 | 0.900 | 0.963 | 0.838 | 0.890 | 0.882 | 0.837 | 0.802 | 0.725 |
| SemgHandMovementCh2 | 0.911 | 0.833 | 0.836 | 0.713 | 0.860 | 0.700 | 0.789 | 0.593 | 0.613 | 0.584 | 0.420 |
| SemgHandSubjectCh2 | 0.958 | 0.945 | 0.924 | 0.813 | 0.951 | 0.813 | 0.853 | 0.771 | 0.753 | 0.727 | 0.484 |
| ShakeGestureWiimoteZ | 0.960 | 0.920 | 0.920 | 0.900 | 0.940 | 0.900 | 0.920 | 0.820 | 0.860 | 0.860 | 0.760 |
| ShapeletSim | 1.000 | 0.856 | 0.856 | 1.000 | 1.000 | 0.911 | 0.672 | 0.589 | 0.683 | 0.650 | 0.489 |
| ShapesAll | 0.908 | 0.855 | 0.852 | 0.855 | 0.902 | 0.840 | 0.848 | 0.788 | 0.773 | 0.768 | 0.733 |
| SmallKitchenAppliances | 0.768 | 0.773 | 0.773 | 0.699 | 0.731 | 0.741 | 0.677 | 0.725 | 0.691 | 0.643 | 0.592 |
| SmoothSubspace | 0.987 | 1.000 | 1.000 | 1.000 | 0.980 | 0.993 | 0.960 | 0.913 | 0.953 | 0.827 | 0.827 |
| SonyAIBORobotSurface1 | 0.965 | 0.921 | 0.927 | 0.918 | 0.903 | 0.912 | 0.902 | 0.804 | 0.899 | 0.725 | 0.724 |
| SonyAIBORobotSurface2 | 0.902 | 0.953 | 0.953 | 0.858 | 0.871 | 0.934 | 0.889 | 0.834 | 0.907 | 0.831 | 0.745 |
| StarLightCurves | 0.975 | 0.973 | 0.973 | 0.979 | 0.969 | 0.972 | 0.964 | 0.968 | 0.967 | 0.907 | 0.949 |
| Strawberry | 0.968 | 0.978 | 0.978 | 0.978 | 0.962 | 0.970 | 0.954 | 0.951 | 0.965 | 0.941 | 0.916 |
| SwedishLeaf | 0.946 | 0.954 | 0.950 | 0.962 | 0.941 | 0.938 | 0.914 | 0.880 | 0.923 | 0.792 | 0.738 |
| Symbols | 0.982 | 0.979 | 0.979 | 0.971 | 0.976 | 0.961 | 0.963 | 0.885 | 0.916 | 0.950 | 0.786 |
| SyntheticControl | 1.000 | 1.000 | 1.000 | 1.000 | 0.997 | 0.993 | 0.987 | 1.000 | 0.990 | 0.993 | 0.490 |
| ToeSegmentation1 | 0.969 | 0.930 | 0.934 | 0.947 | 0.917 | 0.890 | 0.939 | 0.864 | 0.930 | 0.772 | 0.807 |
| ToeSegmentation2 | 0.946 | 0.923 | 0.915 | 0.908 | 0.892 | 0.908 | 0.900 | 0.831 | 0.877 | 0.838 | 0.615 |
| Trace | 1.000 | 1.000 | 1.000 | 1.000 | 1.000 | 1.000 | 0.990 | 1.000 | 1.000 | 1.000 | 1.000 |
| TwoLeadECG | 0.989 | 0.999 | 0.998 | 0.999 | 0.986 | 0.985 | 0.999 | 0.993 | 0.976 | 0.905 | 0.871 |
| TwoPatterns | 1.000 | 1.000 | 1.000 | 1.000 | 1.000 | 0.994 | 0.999 | 1.000 | 0.999 | 1.000 | 0.466 |
| UMD | 1.000 | 1.000 | 1.000 | 1.000 | 1.000 | 1.000 | 0.993 | 0.993 | 0.986 | 0.993 | 0.910 |
| UWaveGestureLibraryAll | 0.947 | 0.966 | 0.967 | 0.878 | 0.930 | 0.956 | 0.896 | 0.903 | 0.692 | 0.892 | 0.475 |
| UWaveGestureLibraryX | 0.823 | 0.820 | 0.819 | 0.823 | 0.795 | 0.814 | 0.785 | 0.781 | 0.733 | 0.728 | 0.569 |
| UWaveGestureLibraryY | 0.736 | 0.745 | 0.736 | 0.762 | 0.719 | 0.736 | 0.710 | 0.697 | 0.641 | 0.634 | 0.348 |
| UWaveGestureLibraryZ | 0.771 | 0.768 | 0.768 | 0.769 | 0.770 | 0.749 | 0.757 | 0.721 | 0.690 | 0.658 | 0.655 |
| Wafer | 0.999 | 0.999 | 0.998 | 0.998 | 0.998 | 0.996 | 0.992 | 0.994 | 0.994 | 0.980 | 0.991 |
| Wine | 0.907 | 0.963 | 0.963 | 0.944 | 0.870 | 0.907 | 0.815 | 0.759 | 0.778 | 0.574 | 0.500 |
| WordSynonyms | 0.723 | 0.715 | 0.704 | 0.699 | 0.676 | 0.705 | 0.691 | 0.630 | 0.531 | 0.649 | 0.422 |
| Worms | 0.818 | 0.766 | 0.753 | 0.792 | 0.701 | 0.779 | 0.727 | 0.623 | 0.753 | 0.584 | 0.455 |
| WormsTwoClass | 0.883 | 0.818 | 0.857 | 0.844 | 0.805 | 0.792 | 0.792 | 0.727 | 0.753 | 0.623 | 0.584 |
| Yoga | 0.903 | 0.937 | 0.869 | 0.872 | 0.887 | 0.834 | 0.837 | 0.812 | 0.791 | 0.837 | 0.830 |
| DodgerLoopDay | 0.675 | 0.675 | 0.675 | 0.613 | 0.562 | 0.613 | - | - | - | 0.500 | 0.200 |
| DodgerLoopGame | 0.920 | 0.971 | 0.942 | 0.913 | 0.841 | 0.739 | - | - | - | 0.877 | 0.696 |
| DodgerLoopWeekend | 0.964 | 0.986 | 0.986 | 0.978 | 0.964 | 0.978 | - | - | - | 0.949 | 0.732 |
| On the first 125 datasets: | | | | | | | | | | | |
| Average Accuracy | 0.864 | 0.841 | 0.838 | 0.832 | 0.830 | 0.823 | 0.806 | 0.761 | 0.757 | 0.727 | 0.641 |
| Average Rank | 2.020 | 2.710 | 3.100 | 3.820 | 4.780 | 4.770 | 6.340 | 7.590 | 7.500 | 8.720 | 9.790 |

Table A3: Performance of TimeHUT on the 30 individual datasets from UEA.

| Dataset | TimeHUT | AutoTCL | SelfDis | InfoTS$_s$ | FEAT | SMDE | InfoTS | TS2Vec | TNC | TS-TCC | T-Loss | DTW | TST | MCL | TF-C |
|---|---|---|---|---|---|---|---|---|---|---|---|---|---|---|---|
| ArticularyWordRecognition | 0.993 | 0.983 | 0.990 | 0.993 | 0.991 | 0.963 | 0.987 | 0.987 | 0.973 | 0.953 | 0.943 | 0.987 | 0.977 | 0.963 | 0.263 |
| AtrialFibrillation | 0.533 | 0.467 | 0.267 | 0.267 | 0.293 | 0.200 | 0.200 | 0.200 | 0.133 | 0.267 | 0.133 | 0.200 | 0.067 | 0.400 | 0.200 |
| BasicMotions | 1.000 | 1.000 | 0.925 | 1.000 | 1.000 | 0.975 | 0.975 | 0.975 | 0.975 | 1.000 | 1.000 | 0.975 | 0.975 | 0.700 | 0.850 |
| CharacterTrajectories | 0.997 | 0.976 | 0.994 | 0.987 | 0.993 | 0.992 | 0.974 | 0.995 | 0.967 | 0.985 | 0.993 | 0.989 | 0.975 | 0.938 | 0.061 |
| Cricket | 1.000 | 1.000 | 1.000 | 1.000 | 0.969 | 1.000 | 0.986 | 0.972 | 0.958 | 0.917 | 0.972 | 1.000 | 1.000 | 0.861 | 0.263 |
| DuckDuckGeese | 0.600 | 0.700 | 0.480 | 0.600 | 0.564 | 0.600 | 0.540 | 0.680 | 0.460 | 0.380 | 0.650 | 0.600 | 0.620 | 0.160 | 0.220 |
| EigenWorms | 0.962 | 0.901 | 0.931 | 0.748 | 0.811 | 0.839 | 0.733 | 0.847 | 0.840 | 0.779 | 0.840 | 0.618 | 0.748 | 0.526 | 0.366 |
| Epilepsy | 0.964 | 0.978 | 0.986 | 0.993 | 0.948 | 0.971 | 0.971 | 0.964 | 0.957 | 0.957 | 0.971 | 0.964 | 0.949 | 0.695 | 0.826 |
| ERing | 0.926 | 0.944 | 0.919 | 0.953 | 0.896 | 0.900 | 0.949 | 0.874 | 0.852 | 0.904 | 0.133 | 0.133 | 0.874 | 0.825 | 0.500 |
| EthanolConcentration | 0.373 | 0.354 | 0.460 | 0.323 | 0.322 | 0.289 | 0.281 | 0.308 | 0.297 | 0.285 | 0.205 | 0.323 | 0.262 | 0.251 | 0.251 |
| FaceDetection | 0.551 | 0.581 | 0.541 | 0.525 | 0.530 | 0.548 | 0.534 | 0.501 | 0.536 | 0.544 | 0.513 | 0.529 | 0.534 | 0.517 | 0.510 |
| FingerMovements | 0.642 | 0.640 | 0.590 | 0.620 | 0.488 | 0.530 | 0.630 | 0.480 | 0.470 | 0.460 | 0.580 | 0.530 | 0.560 | 0.530 | 0.530 |
| HandMovementDirection | 0.432 | 0.432 | 0.432 | 0.514 | 0.378 | 0.378 | 0.392 | 0.338 | 0.324 | 0.243 | 0.351 | 0.231 | 0.243 | 0.283 | 0.216 |
| Handwriting | 0.582 | 0.384 | 0.428 | 0.554 | 0.542 | 0.418 | 0.452 | 0.515 | 0.249 | 0.498 | 0.451 | 0.286 | 0.225 | 0.308 | 0.137 |
| Heartbeat | 0.810 | 0.785 | 0.751 | 0.771 | 0.746 | 0.746 | 0.722 | 0.683 | 0.746 | 0.751 | 0.741 | 0.717 | 0.746 | 0.634 | 0.722 |
| JapaneseVowels | 0.984 | 0.984 | 0.978 | 0.986 | 0.983 | 0.970 | 0.984 | 0.984 | 0.978 | 0.930 | 0.989 | 0.949 | 0.978 | 0.875 | 0.408 |
| Libras | 0.883 | 0.833 | 0.900 | 0.889 | 0.889 | 0.850 | 0.883 | 0.867 | 0.817 | 0.822 | 0.883 | 0.870 | 0.656 | 0.655 | 0.438 |
| LSST | 0.586 | 0.554 | 0.640 | 0.593 | 0.548 | 0.618 | 0.591 | 0.537 | 0.595 | 0.474 | 0.509 | 0.551 | 0.408 | 0.404 | 0.223 |
| MotorImagery | 0.660 | 0.570 | 0.520 | 0.610 | 0.562 | 0.620 | 0.630 | 0.510 | 0.500 | 0.610 | 0.580 | 0.500 | 0.500 | 0.480 | 0.500 |
| NATOPS | 0.966 | 0.944 | 0.972 | 0.939 | 0.921 | 0.916 | 0.933 | 0.928 | 0.911 | 0.822 | 0.917 | 0.883 | 0.850 | 0.733 | 0.550 |
| PEMS-SF | 0.855 | 0.838 | 0.884 | 0.757 | 0.874 | 0.809 | 0.751 | 0.682 | 0.699 | 0.734 | 0.676 | 0.711 | 0.740 | 0.849 | 0.346 |
| PenDigits | 0.989 | 0.984 | 0.987 | 0.989 | 0.989 | 0.981 | 0.990 | 0.989 | 0.979 | 0.974 | 0.981 | 0.977 | 0.560 | 0.973 | 0.708 |
| PhonemeSpectra | 0.243 | 0.218 | 0.292 | 0.233 | 0.216 | 0.218 | 0.249 | 0.233 | 0.207 | 0.252 | 0.222 | 0.151 | 0.085 | 0.102 | 0.292 |
| RacketSports | 0.921 | 0.914 | 0.908 | 0.829 | 0.888 | 0.842 | 0.855 | 0.855 | 0.776 | 0.816 | 0.855 | 0.803 | 0.809 | 0.776 | 0.644 |
| SelfRegulationSCP1 | 0.860 | 0.891 | 0.860 | 0.887 | 0.852 | 0.894 | 0.874 | 0.812 | 0.799 | 0.823 | 0.843 | 0.775 | 0.754 | 0.774 | 0.658 |
| SelfRegulationSCP2 | 0.583 | 0.578 | 0.600 | 0.572 | 0.562 | 0.577 | 0.578 | 0.578 | 0.550 | 0.533 | 0.539 | 0.539 | 0.550 | 0.505 | 0.500 |
| SpokenArabicDigits | 0.995 | 0.925 | 0.992 | 0.932 | 0.986 | 0.977 | 0.947 | 0.988 | 0.934 | 0.970 | 0.905 | 0.963 | 0.923 | 0.951 | 0.104 |
| StandWalkJump | 0.600 | 0.533 | 0.533 | 0.467 | 0.533 | 0.533 | 0.467 | 0.467 | 0.400 | 0.333 | 0.333 | 0.200 | 0.267 | 0.266 | 0.333 |
| UWaveGestureLibrary | 0.916 | 0.893 | 0.919 | 0.884 | 0.929 | 0.918 | 0.884 | 0.906 | 0.759 | 0.753 | 0.875 | 0.903 | 0.575 | 0.825 | 0.565 |
| InsectWingbeat | 0.466 | 0.488 | 0.449 | 0.472 | 0.462 | 0.553 | 0.470 | 0.466 | 0.469 | 0.264 | 0.156 | - | 0.105 | 0.275 | 0.100 |
| On the first 29 datasets: | | | | | | | | | | | | | | | |
| Average Accuracy | 0.773 | 0.751 | 0.748 | 0.738 | 0.731 | 0.727 | 0.722 | 0.712 | 0.677 | 0.682 | 0.675 | 0.650 | 0.635 | 0.612 | 0.420 |
| Average Rank | 2.93 | 4.90 | 4.24 | 5.07 | 6.40 | 6.74 | 6.60 | 7.55 | 10.38 | 9.47 | 8.76 | 10.02 | 10.97 | 12.45 | 13.53 |

Table A4: Forecasting Performance on ETTh1 and ETTh2 (MSE)

| Dataset | H | TimeHUT | TS2Vec | Informer | LogTrans | N-BEATS | TCN | TS-TCC | LSTnet |
|---|---|---|---|---|---|---|---|---|---|
| | 24 | **0.037** | 0.039 | 0.098 | 0.103 | 0.094 | 0.075 | 0.103 | 0.108 |
| | 48 | **0.054** | 0.062 | 0.158 | 0.167 | 0.210 | 0.227 | 0.139 | 0.175 |
| ETTh$_1$ | 168 | **0.098** | 0.134 | 0.183 | 0.207 | 0.232 | 0.316 | 0.253 | 0.396 |
| | 336 | **0.115** | 0.154 | 0.222 | 0.230 | 0.306 | 0.468 | 0.155 | 0.468 |
| | 720 | **0.138** | 0.163 | 0.269 | 0.273 | 0.322 | 0.390 | 0.190 | 0.659 |
| | 24 | **0.090** | 0.090 | 0.093 | 0.102 | 0.198 | 0.103 | 0.239 | 3.554 |
| | 48 | **0.122** | 0.124 | 0.155 | 0.169 | 0.234 | 0.142 | 0.260 | 3.190 |
| ETTh$_2$ | 168 | **0.188** | 0.208 | 0.232 | 0.246 | 0.331 | 0.227 | 0.291 | 2.800 |
| | 336 | **0.195** | 0.213 | 0.263 | 0.267 | 0.431 | 0.296 | 0.336 | 2.753 |
| | 720 | **0.206** | 0.214 | 0.277 | 0.303 | 0.437 | 0.325 | 0.362 | 2.878 |
| Average | | **0.125** | 0.136 | 0.198 | 0.213 | 0.290 | 0.227 | 0.234 | 1.299 |

Table A5: Hyperparameters for the 128 individual datasets of UCR (Part 1).

| Dataset | TimeHUT Acc | AUPRC | $c_i/c_t/m_a/\tau_{min}/\tau_{max}/T_{max}$ |
|---|---|---|---|
| ACSF1 | 0.920 | 0.933 | 0.0 / 0.0 / 0.76 / 0.4 / 0.66 / 19 |
| Adiac | 0.783 | 0.774 | 0.47 / 2.68 / 0.7 / 0.01 / 0.62 / 10 |
| AllGestureWiimoteX | 0.796 | 0.790 | 10.0 / 0.48 / 0.77 / 0.2 / 0.77 / 32 |
| AllGestureWiimoteY | 0.813 | 0.817 | 7.99 / 0.0 / 0.61 / 0.32 / 0.5 / 21 |
| AllGestureWiimoteZ | 0.793 | 0.796 | 7.61 / 2.09 / 0.8 / 0.32 / 0.94 / 13 |
| ArrowHead | 0.863 | 0.914 | 2.34 / 1.27 / 0.76 / 0.0 / 0.62 / 46 |
| BME | 1.000 | 1.000 | 8.17 / 4.17 / 0.65 / 0.01 / 0.75 / 20 |
| Beef | 0.833 | 0.869 | 3.01 / 0.46 / 0.49 / 0.21 / 0.86 / 16 |
| BeetleFly | 0.900 | 0.971 | 8.69 / 7.46 / 0.76 / 0.09 / 0.91 / 26 |
| BirdChicken | 1.000 | 1.000 | 0.47 / 1.48 / 0.64 / 0.09 / 0.56 / 37 |
| CBF | 1.000 | 1.000 | 4.52 / 6.89 / 0.57 / 0.0 / 0.94 / 46 |
| Car | 0.933 | 0.935 | 0.0 / 0.24 / 0.8 / 0.2 / 0.56 / 17 |
| Chinatown | 0.985 | 0.999 | 10.0 / 7.53 / 0.3 / 0.05 / 0.76 / 25 |
| ChlorineConcentration | 0.842 | 0.827 | 1.28 / 1.24 / 0.69 / 0.15 / 0.61 / 24 |
| CinCECGTorso | 0.848 | 0.910 | 5.28 / 7.62 / 0.44 / 0.04 / 0.57 / 10 |
| Coffee | 1.000 | 1.000 | 5.34 / 8.51 / 0.55 / 0.3 / 0.88 / 37 |
| Computers | 0.716 | 0.756 | 8.29 / 2.26 / 0.77 / 0.22 / 0.74 / 14 |
| CricketX | 0.821 | 0.829 | 0.0 / 10.0 / 0.65 / 0.17 / 0.6 / 38 |
| CricketY | 0.800 | 0.801 | 3.87 / 7.79 / 0.73 / 0.12 / 0.96 / 10 |
| CricketZ | 0.818 | 0.829 | 10.0 / 0.87 / 0.79 / 0.31 / 0.77 / 39 |
| Crop | 0.768 | 0.761 | 0.0 / 10.0 / 0.8 / 0.3 / 0.63 / 42 |
| DiatomSizeReduction | 0.990 | 0.983 | 9.89 / 2.68 / 0.64 / 0.21 / 0.97 / 12 |
| DistalPhalanxOutlineAgeGroup | 0.734 | 0.744 | 1.72 / 9.06 / 0.52 / 0.23 / 0.82 / 18 |
| DistalPhalanxOutlineCorrect | 0.793 | 0.841 | 5.15 / 2.37 / 0.31 / 0.14 / 0.61 / 29 |
| DistalPhalanxTW | 0.734 | 0.550 | 8.45 / 1.46 / 0.67 / 0.4 / 0.72 / 47 |
| DodgerLoopDay | 0.675 | 0.653 | 8.34 / 9.28 / 0.64 / 0.28 / 1.0 / 28 |
| DodgerLoopGame | 0.920 | 0.974 | 8.67 / 8.14 / 0.63 / 0.37 / 0.82 / 28 |
| DodgerLoopWeekend | 0.964 | 0.981 | 1.55 / 8.77 / 0.42 / 0.0 / 0.86 / 45 |
| ECG200 | 0.930 | 0.982 | 2.48 / 5.82 / 0.39 / 0.1 / 0.5 / 37 |
| ECG5000 | 0.945 | 0.561 | 0.0 / 10.0 / 0.45 / 0.08 / 1.0 / 18 |
| ECGFiveDays | 1.000 | 1.000 | 5.98 / 7.08 / 0.51 / 0.27 / 0.66 / 47 |
| EOGHorizontalSignal | 0.594 | 0.625 | 5.48 / 1.64 / 0.62 / 0.22 / 0.54 / 17 |
| EOGVerticalSignal | 0.533 | 0.513 | 2.1 / 1.83 / 0.59 / 0.03 / 0.83 / 10 |
| Earthquakes | 0.827 | 0.622 | 4.31 / 1.67 / 0.46 / 0.02 / 0.77 / 13 |
| ElectricDevices | 0.745 | 0.650 | 0.84 / 0.0 / 0.3 / 0.16 / 0.57 / 18 |
| EthanolLevel | 0.588 | 0.559 | 0.16 / 1.36 / 0.65 / 0.08 / 0.51 / 13 |
| FaceAll | 0.914 | 0.905 | 10.0 / 0.0 / 0.41 / 0.15 / 0.5 / 50 |
| FaceFour | 0.955 | 0.990 | 6.0 / 5.73 / 0.62 / 0.03 / 0.84 / 22 |
| FacesUCR | 0.945 | 0.945 | 0.0 / 3.54 / 0.7 / 0.05 / 0.56 / 36 |
| FiftyWords | 0.802 | 0.722 | 2.02 / 6.94 / 0.32 / 0.39 / 0.59 / 47 |
| Fish | 0.966 | 0.980 | 4.12 / 10.0 / 0.74 / 0.1 / 0.5 / 10 |
| FordA | 0.936 | 0.981 | 0.94 / 0.0 / 0.3 / 0.33 / 0.64 / 29 |
| FordB | 0.817 | 0.898 | 1.8 / 1.97 / 0.33 / 0.23 / 0.74 / 10 |
| FreezerRegularTrain | 0.994 | 0.996 | 1.32 / 4.69 / 0.76 / 0.08 / 0.71 / 17 |
| FreezerSmallTrain | 0.988 | 0.992 | 1.73 / 5.79 / 0.45 / 0.4 / 0.65 / 35 |
| Fungi | 1.000 | 1.000 | 1.85 / 3.17 / 0.69 / 0.1 / 0.67 / 29 |
| GestureMidAirD1 | 0.700 | 0.719 | 0.0 / 7.82 / 0.32 / 0.31 / 0.53 / 24 |
| GestureMidAirD2 | 0.608 | 0.672 | 0.39 / 6.08 / 0.47 / 0.11 / 0.62 / 31 |
| GestureMidAirD3 | 0.438 | 0.503 | 0.56 / 3.72 / 0.79 / 0.08 / 0.83 / 22 |
| GesturePebbleZ1 | 0.913 | 0.969 | 0.0 / 0.55 / 0.8 / 0.06 / 0.99 / 38 |
| GesturePebbleZ2 | 0.943 | 0.960 | 4.98 / 2.3 / 0.55 / 0.16 / 0.69 / 23 |
| GunPoint | 0.993 | 0.999 | 9.18 / 0.71 / 0.54 / 0.34 / 0.72 / 35 |
| GunPointAgeSpan | 0.997 | 1.000 | 0.0 / 10.0 / 0.45 / 0.09 / 0.78 / 50 |
| GunPointMaleVersusFemale | 1.000 | 1.000 | 5.94 / 2.46 / 0.59 / 0.16 / 0.77 / 50 |
| GunPointOldVersusYoung | 1.000 | 1.000 | 2.16 / 8.84 / 0.75 / 0.26 / 0.64 / 38 |
| Ham | 0.781 | 0.878 | 4.98 / 6.76 / 0.72 / 0.04 / 0.8 / 11 |
| HandOutlines | 0.946 | 0.966 | 8.79 / 1.91 / 0.4 / 0.09 / 0.7 / 10 |
| Haptics | 0.549 | 0.563 | 0.0 / 3.22 / 0.46 / 0.07 / 0.74 / 45 |
| Herring | 0.703 | 0.613 | 6.22 / 4.47 / 0.62 / 0.0 / 0.5 / 26 |
| HouseTwenty | 0.983 | 0.985 | 0.12 / 0.0 / 0.34 / 0.08 / 0.53 / 50 |
| InlineSkate | 0.456 | 0.423 | 7.85 / 10.0 / 0.3 / 0.1 / 0.63 / 45 |
| InsectEPGRegularTrain | 1.000 | 1.000 | 9.19 / 4.05 / 0.55 / 0.33 / 0.94 / 46 |
| InsectEPGSmallTrain | 1.000 | 1.000 | 3.0 / 3.0 / 0.5 / 0.07 / 0.8 / 30 |
| InsectWingbeatSound | 0.644 | 0.659 | 4.61 / 5.35 / 0.77 / 0.29 / 0.5 / 49 |

Table A6: Hyperparameters for the 128 individual datasets of UCR (Part 2).

| Dataset | TimeHUT Acc | AUPRC | $c_i/c_t/m_a/\tau_{min}/\ \tau_{max}/T_{max}$ |
|---|---|---|---|
| ItalyPowerDemand | 0.971 | 0.974 | 7.39 / 10.0 / 0.65 / 0.03 / 0.99 / 11 |
| LargeKitchenAppliances | 0.885 | 0.900 | 5.74 / 0.7 / 0.58 / 0.4 / 0.85 / 11 |
| Lightning2 | 0.934 | 0.978 | 1.41 / 6.16 / 0.34 / 0.01 / 0.6 / 14 |
| Lightning7 | 0.877 | 0.927 | 9.64 / 4.48 / 0.58 / 0.22 / 0.99 / 30 |
| Mallat | 0.969 | 0.983 | 0.44 / 10.0 / 0.66 / 0.02 / 0.7 / 18 |
| Meat | 0.967 | 1.000 | 9.33 / 5.65 / 0.66 / 0.07 / 0.92 / 16 |
| MedicalImages | 0.828 | 0.816 | 0.72 / 0.0 / 0.64 / 0.03 / 0.58 / 10 |
| MelbournePedestrian | 0.964 | 0.962 | 10.0 / 1.92 / 0.77 / 0.23 / 0.64 / 33 |
| MiddlePhalanxOutlineAgeGroup | 0.662 | 0.532 | 2.12 / 9.61 / 0.8 / 0.12 / 0.74 / 26 |
| MiddlePhalanxOutlineCorrect | 0.869 | 0.924 | 2.48 / 6.81 / 0.6 / 0.21 / 1.0 / 38 |
| MiddlePhalanxTW | 0.617 | 0.423 | 1.57 / 5.34 / 0.8 / 0.21 / 0.74 / 17 |
| MixedShapesRegularTrain | 0.929 | 0.952 | 5.1 / 8.41 / 0.33 / 0.24 / 0.9 / 24 |
| MixedShapesSmallTrain | 0.894 | 0.927 | 4.95 / 9.89 / 0.31 / 0.4 / 0.91 / 50 |
| MoteStrain | 0.925 | 0.967 | 4.43 / 9.69 / 0.75 / 0.03 / 0.83 / 22 |
| NonInvasiveFetalECGThorax1 | 0.948 | 0.946 | 1.1 / 9.14 / 0.57 / 0.21 / 0.54 / 48 |
| NonInvasiveFetalECGThorax2 | 0.953 | 0.951 | 0.0 / 1.92 / 0.78 / 0.02 / 0.74 / 26 |
| OSULeaf | 0.864 | 0.887 | 2.96 / 3.95 / 0.69 / 0.02 / 0.58 / 32 |
| OliveOil | 0.933 | 0.910 | 5.19 / 5.78 / 0.62 / 0.09 / 0.67 / 30 |
| PLAID | 0.553 | 0.523 | 0.0 / 0.0 / 0.8 / 0.02 / 0.93 / 50 |
| PhalangesOutlinesCorrect | 0.819 | 0.897 | 0.54 / 1.3 / 0.66 / 0.36 / 0.55 / 41 |
| Phoneme | 0.320 | 0.146 | 0.0 / 0.06 / 0.8 / 0.01 / 0.69 / 50 |
| PickupGestureWiimoteZ | 0.940 | 0.947 | 7.53 / 6.34 / 0.76 / 0.29 / 0.86 / 43 |
| PigAirwayPressure | 0.817 | 0.838 | 9.95 / 0.0 / 0.7 / 0.35 / 1.0 / 18 |
| PigArtPressure | 0.971 | 0.981 | 5.86 / 0.03 / 0.62 / 0.25 / 0.76 / 35 |
| PigCVP | 0.928 | 0.936 | 5.92 / 0.35 / 0.8 / 0.18 / 0.96 / 22 |
| Plane | 1.000 | 1.000 | 7.08 / 0.06 / 0.41 / 0.14 / 0.57 / 17 |
| PowerCons | 1.000 | 1.000 | 0.0 / 7.37 / 0.37 / 0.33 / 0.5 / 32 |
| ProximalPhalanxOutlineAgeGroup | 0.859 | 0.713 | 2.44 / 0.63 / 0.46 / 0.26 / 0.56 / 17 |
| ProximalPhalanxOutlineCorrect | 0.914 | 0.964 | 7.67 / 10.0 / 0.57 / 0.17 / 0.64 / 39 |
| ProximalPhalanxTW | 0.849 | 0.631 | 4.34 / 2.33 / 0.55 / 0.1 / 0.69 / 28 |
| RefrigerationDevices | 0.645 | 0.630 | 0.0 / 0.76 / 0.74 / 0.0 / 0.85 / 21 |
| Rock | 0.820 | 0.804 | 4.31 / 0.49 / 0.54 / 0.19 / 0.78 / 17 |
| ScreenType | 0.403 | 0.386 | 1.62 / 3.41 / 0.39 / 0.26 / 0.73 / 41 |
| SemgHandGenderCh2 | 0.973 | 0.991 | 6.46 / 5.33 / 0.3 / 0.18 / 0.88 / 30 |
| SemgHandMovementCh2 | 0.911 | 0.921 | 4.61 / 10.0 / 0.38 / 0.29 / 0.73 / 29 |
| SemgHandSubjectCh2 | 0.958 | 0.963 | 4.68 / 2.35 / 0.41 / 0.39 / 0.84 / 20 |
| ShakeGestureWiimoteZ | 0.960 | 0.971 | 4.54 / 7.47 / 0.58 / 0.26 / 0.75 / 38 |
| ShapeletSim | 1.000 | 1.000 | 3.87 / 0.36 / 0.65 / 0.12 / 0.74 / 16 |
| ShapesAll | 0.908 | 0.922 | 10.0 / 3.7 / 0.4 / 0.05 / 0.5 / 37 |
| SmallKitchenAppliances | 0.768 | 0.825 | 10.0 / 0.0 / 0.75 / 0.14 / 0.79 / 10 |
| SmoothSubspace | 0.987 | 0.995 | 5.2 / 0.36 / 0.42 / 0.08 / 0.84 / 11 |
| SonyAIBORobotSurface1 | 0.965 | 0.996 | 5.71 / 6.48 / 0.8 / 0.02 / 0.99 / 10 |
| SonyAIBORobotSurface2 | 0.902 | 0.985 | 4.16 / 10.0 / 0.8 / 0.01 / 0.53 / 45 |
| StarLightCurves | 0.975 | 0.982 | 10.0 / 0.2 / 0.79 / 0.21 / 1.0 / 33 |
| Strawberry | 0.968 | 0.997 | 0.0 / 10.0 / 0.5 / 0.28 / 0.5 / 37 |
| SwedishLeaf | 0.946 | 0.944 | 2.55 / 6.61 / 0.76 / 0.37 / 0.51 / 16 |
| Symbols | 0.982 | 0.989 | 6.38 / 3.52 / 0.47 / 0.4 / 0.73 / 46 |
| SyntheticControl | 1.000 | 1.000 | 0.25 / 9.78 / 0.33 / 0.14 / 0.74 / 43 |
| ToeSegmentation1 | 0.969 | 0.988 | 10.0 / 6.39 / 0.69 / 0.4 / 0.95 / 50 |
| ToeSegmentation2 | 0.946 | 0.944 | 5.23 / 6.56 / 0.66 / 0.29 / 0.92 / 11 |
| Trace | 1.000 | 1.000 | 9.03 / 6.29 / 0.33 / 0.24 / 0.64 / 26 |
| TwoLeadECG | 0.989 | 1.000 | 0.0 / 9.41 / 0.8 / 0.2 / 0.5 / 10 |
| TwoPatterns | 1.000 | 1.000 | 3.0 / 3.0 / 0.5 / 0.07 / 0.8 / 30 |
| UMD | 1.000 | 1.000 | 9.75 / 5.93 / 0.44 / 0.08 / 0.88 / 12 |
| UWaveGestureLibraryAll | 0.947 | 0.966 | 5.33 / 4.61 / 0.3 / 0.09 / 1.0 / 18 |
| UWaveGestureLibraryX | 0.823 | 0.820 | 1.59 / 6.69 / 0.42 / 0.08 / 0.8 / 28 |
| UWaveGestureLibraryY | 0.736 | 0.734 | 3.14 / 8.18 / 0.44 / 0.28 / 1.0 / 35 |
| UWaveGestureLibraryZ | 0.771 | 0.778 | 8.33 / 0.23 / 0.8 / 0.2 / 0.76 / 10 |
| Wafer | 0.999 | 1.000 | 1.98 / 1.83 / 0.62 / 0.02 / 0.77 / 38 |
| Wine | 0.907 | 0.970 | 4.31 / 8.83 / 0.71 / 0.08 / 0.9 / 14 |
| WordSynonyms | 0.723 | 0.585 | 2.56 / 6.52 / 0.76 / 0.17 / 0.5 / 48 |
| Worms | 0.818 | 0.828 | 3.61 / 0.0 / 0.75 / 0.21 / 1.0 / 41 |
| WormsTwoClass | 0.883 | 0.873 | 0.76 / 2.79 / 0.34 / 0.21 / 0.68 / 50 |
| Yoga | 0.903 | 0.924 | 8.48 / 0.53 / 0.52 / 0.39 / 0.68 / 10 |
| Average on all 128 datasets | 0.864 | 0.862 | |

Table A7: Hyperparameters for the 30 individual datasets of UEA.

| Dataset | TimeHUT | | $c_i/c_t/m_a/\tau_{min}/\ \tau_{max}/T_{max}$ |
|---|---|---|---|
| | Acc | AUPRC | |
| ArticularyWordRecognition | 0.993 | 0.994 | 8.73 / 8.06 / 0.6 / 0.36 / 0.64 / 37 |
| AtrialFibrillation | 0.533 | 0.440 | 2.04 / 0.08 / 0.67 / 0.26 / 0.68 / 49 |
| BasicMotions | 1.000 | 1.000 | 3.0 / 3.0 / 0.5 / 0.07 / 0.8 / 30 |
| CharacterTrajectories | 0.997 | 0.998 | 0.32 / 1.1 / 0.6 / 0.26 / 1.0 / 20 |
| Cricket | 1.000 | 1.000 | 5.24 / 9.16 / 0.37 / 0.33 / 0.87 / 14 |
| DuckDuckGeese | 0.600 | 0.630 | 0.1 / 1.0 / 0.5 / 0.0 / 1.0 / 1.0 |
| ERing | 0.926 | 0.969 | 0.0 / 0.75 / 0.68 / 0.0 / 1.0 / 17 |
| EigenWorms | 0.962 | 0.951 | 10.0 / 5.15 / 0.51 / 0.2 / 0.91 / 28 |
| Epilepsy | 0.964 | 0.973 | 0.41 / 0.12 / 0.52 / 0.28 / 0.6 / 34 |
| EthanolConcentration | 0.373 | 0.361 | 0.78 / 4.31 / 0.52 / 0.11 / 0.5 / 47 |
| FaceDetection | 0.551 | 0.551 | 9.04 / 10.0 / 0.74 / 0.38 / 0.66 / 28 |
| FingerMovements | 0.642 | 0.652 | 0.0 / 0.0 / 0.5 / 0.1 / 0.6 / 50 |
| HandMovementDirection | 0.432 | 0.461 | 3.42 / 0.0 / 0.38 / 0.18 / 0.67 / 10 |
| Handwriting | 0.582 | 0.586 | 4.51 / 0.0 / 0.8 / 0.07 / 0.62 / 17 |
| Heartbeat | 0.810 | 0.586 | 1.64 / 5.66 / 0.46 / 0.22 / 0.66 / 33 |
| InsectWingbeat | 0.466 | 0.453 | 0.0 / 0.0 / 0.3 / 0.24 / 0.99 / 21 |
| JapaneseVowels | 0.984 | 0.996 | 9.33 / 8.55 / 0.47 / 0.33 / 0.68 / 33 |
| LSST | 0.586 | 0.407 | 2.59 / 0.0 / 0.34 / 0.16 / 0.88 / 22 |
| Libras | 0.883 | 0.892 | 7.76 / 5.46 / 0.68 / 0.09 / 0.72 / 20 |
| MotorImagery | 0.660 | 0.661 | 0.5/ 0.2 / 0.5 / 0.0 / 1.0 / 1.0 |
| NATOPS | 0.966 | 0.985 | 0.0 / 0.0 / 0.5 / 0.1 / 0.4 / 50 |
| PEMS-SF | 0.855 | 0.894 | 8.8 / 9.3 / 0.6 / 0.24 / 0.98 / 25 |
| PenDigits | 0.989 | 0.995 | 2.8 / 5.59 / 0.67 / 0.05 / 0.92 / 26 |
| PhonemeSpectra | 0.243 | 0.195 | 7.18 / 3.93 / 0.8 / 0.35 / 0.5 / 22 |
| RacketSports | 0.921 | 0.905 | 0.82 / 4.83 / 0.57 / 0.09 / 0.63 / 48 |
| SelfRegulationSCP1 | 0.860 | 0.935 | 0.95 / 0.0 / 0.49 / 0.0 / 0.53 / 43 |
| SelfRegulationSCP2 | 0.583 | 0.566 | 0.0 / 4.06 / 0.35 / 0.02 / 0.5 / 21 |
| SpokenArabicDigits | 0.995 | 0.998 | 8.45 / 8.92 / 0.49 / 0.15 / 0.94 / 10 |
| StandWalkJump | 0.600 | 0.437 | 0.2 / 0.2 / 0.5 / 0.0 / 1.0 / 1.0 |
| UWaveGestureLibrary | 0.916 | 0.947 | 0.12 / 0.46 / 0.8 / 0.08 / 0.95 / 13 |
| Average on all 30 datasets: | 0.762 | 0.747 | |

Table A8: Hyperparameters for anomaly detection.

| Dataset | F1 | Precision | Recall | $c_i/c_t/m_a/\tau_{min}/\tau_{max}/T_{max}$ |
|---|---|---|---|---|
| Yahoo | 0.755 | 0.746 | 0.764 | 9.9 / 0.02 / 0.56 / 0.31 / 1.0 / 50.0 |
| KPI | 0.721 | 0.899 | 0.602 | 4.4 / 0.08 / 0.50 / 0.24 / 0.5 / 44.0 |
| *Cold-start*: | | | | |
| Yahoo | 0.779 | 0.793 | 0.765 | 10.00 / 6.27 / 0.76 / 0.03 / 0.99 / 39 |
| KPI | 0.691 | 0.894 | 0.563 | 2.35 / 0.87 / 0.62 / 0.32 / 0.55 / 47 |

Algorithm 1: PyTorch-like pseudo-code for TimeHUT.

```
1   # B: Batch size
2   # T: Length of the time-series
3   # tau_min, tau_max, omega, sigma: Temperature scheduling parameters
4   # m_a: Angular margin
5
6   Input: Time-series dataset X = [x1, x2,..., x_i,..., xn]
7   Model parameters \theta
8   Output: Learned representations Z = [z1, z2,..., z_i,..., zn]
9
10  # Step 1: Preprocessing
11  for x_i in X:
12      a1, b1, a2, b2 = random_crop_indices(x_i) # Randomly crop indices for subseries 1 and 2
13      x1[i] = x_i[a1:b1] # First subseries
14      x2[i] = x_i[a2:b2] # Second subseries
15      z1[i] = encoder(x1[i]) # Encode first subseries
16      z2[i] = encoder(x2[i]) # Encode second subseries
17
18  # Step 2: Hierarchical Contrastive Loss Calculation
19  for t, t' in overlapping_timestamps(z1[i], z2[I]): # Iterate over timestamps of instance i
20      s(it_i't) = sim(z1[i][t], z2[i][t]) # Positive pairs
21      s(it_i't') = sim(z1[i][t], z2[i][t']) # Negative pairs across subseries
22      s(it_it') = sim(z1[i][t], z1[i][t']) # Negative pairs within subseries
23
24      L_Temp = temporal_contrastive_loss(s(it_i't), s(it_i't'), s(it_it'))
25  for i in range(B): # Iterate over all instances in the batch
26      s(it_i't) = sim(z1[i][t], z2[i][t]) # Positive pairs
27      s(it_j't) = sim(z1[i][t], z2[j'][t]) # Negative pairs within the batch (cropped)
28      s(it_jt) = sim(z1[i][t], z1[j][t]) # Negative pairs within the batch
29      L_Inst = instance_contrastive_loss(s(it_i't), s(it_j't), s(it_jt))
30
31  # Step 3: Temperature Scheduling
32  tau_sigma = lambda sigma: (tau_max-tau_min) * cos^2(omega * sigma / 2) + tau_min
33  for embedding_pair in embedding_pairs(z1, z2):
34      L_TempSch = L_temp / tau_sigma(sigma)
35      L_InstSch = L_inst / tau_sigma(sigma)
36
37  # Step 4: Hierarchical Angular Margin Loss Calculation
38  for t, t' in overlapping_timestamps(z1[i], z2[i]): # Iterate over timestamps of instance i
39      (\cos^{-1} (s_{it,i't}))^2 # Positive pairs
40      \max(0, m_a - \cos^{-1}(s_{it,i't}))^2 # Negative pairs across subseries
41      \max(0, m_a - \cos^{-1}(s_{it,it'}))^2 # Negative pairs within subseries
42      L_TempAng = compute_angular_margin_loss(s(it_i't), s(it_i't'), s(it_it'), m_a)
43
44  for i in range(B): # Iterate over all instances in the batch
45      (\cos^{-1}(s_{it,i't}))^2 # Positive pairs
46      \max(0, m_a - \cos^{-1}(s_{it,jt}))^2 # Negative pairs within the batch (cropped)
47      \max(0, m_a - \cos^{-1}(s_{it,j't}))^2 # Negative pairs within the batch
48      L_InstAng = compute_angular_margin_loss(s(it_i't), s(it_j't), s(it_jt), m_a)
49
50  # Step 5: Combine Losses
51  L_HierSch = sum(L_temp_sch + L_inst_sch) / (len(X) * T)
52  L_HierAng = sum(L_temp_ang + L_inst_ang) / (len(X) * T)
53  L_Total = L_HierSch + L_HierAng
54
55  # Step 6: Model Optimization
56  theta = optimize(theta, L_Total)
```

Table A9: Mean Difference (MD) values for TimeHUT and competing methods.

| | TimeHUT | AutoTCL | SelfDis | InfoTS$_s$ | FEAT | SMDE | InfoTS | TS2Vec | TNC | TS-TCC | T-Loss | TST | MCL | TF-C |
|---|---|---|---|---|---|---|---|---|---|---|---|---|---|---|
| **TimeHUT** | 0.00 | 0.02 | 0.02 | 0.03 | 0.04 | 0.04 | 0.05 | 0.06 | 0.09 | 0.09 | 0.10 | 0.15 | 0.16 | 0.35 |
| **AutoTCL** | -0.02 | 0.00 | 0.00 | 0.01 | 0.02 | 0.02 | 0.03 | 0.04 | 0.07 | 0.07 | 0.08 | 0.13 | 0.14 | 0.33 |
| **SelfDis** | -0.02 | -0.00 | 0.00 | 0.01 | 0.02 | 0.02 | 0.03 | 0.04 | 0.07 | 0.08 | 0.12 | 0.14 | 0.14 | 0.33 |
| **InfoTS$_s$** | -0.03 | -0.01 | -0.01 | 0.00 | 0.01 | 0.02 | 0.03 | 0.03 | 0.06 | 0.06 | 0.07 | 0.11 | 0.13 | 0.32 |
| **FEAT** | -0.04 | -0.02 | -0.02 | -0.01 | 0.00 | 0.01 | 0.02 | 0.05 | 0.05 | 0.06 | 0.10 | 0.12 | 0.12 | 0.31 |
| **SMDE** | -0.04 | -0.02 | -0.02 | -0.01 | -0.01 | 0.00 | 0.01 | 0.01 | 0.02 | 0.05 | 0.06 | 0.10 | 0.12 | 0.31 |
| **InfoTS** | -0.05 | -0.03 | -0.02 | -0.02 | -0.01 | -0.01 | 0.00 | 0.01 | 0.04 | 0.05 | 0.06 | 0.10 | 0.11 | 0.30 |
| **TS2Vec** | -0.06 | -0.04 | -0.03 | -0.03 | -0.02 | -0.01 | -0.01 | 0.00 | 0.03 | 0.04 | 0.05 | 0.09 | 0.10 | 0.29 |
| **TNC** | -0.09 | -0.07 | -0.07 | -0.06 | -0.05 | -0.04 | -0.03 | -0.03 | 0.00 | -0.00 | 0.01 | 0.05 | 0.07 | 0.26 |
| **TS-TCC** | -0.09 | -0.07 | -0.07 | -0.06 | -0.05 | -0.04 | -0.04 | -0.04 | -0.00 | 0.00 | 0.01 | 0.07 | 0.07 | 0.26 |
| **T-Loss** | -0.10 | -0.08 | -0.08 | -0.06 | -0.06 | -0.06 | -0.05 | -0.05 | -0.01 | -0.00 | 0.00 | 0.04 | 0.06 | 0.25 |
| **TST** | -0.15 | -0.13 | -0.12 | -0.11 | -0.10 | -0.09 | -0.09 | -0.10 | -0.05 | -0.05 | -0.04 | 0.00 | 0.02 | 0.21 |
| **MCL** | -0.16 | -0.14 | -0.14 | -0.13 | -0.12 | -0.11 | -0.10 | -0.11 | -0.07 | -0.07 | -0.06 | -0.02 | 0.00 | 0.19 |
| **TF-C** | -0.35 | -0.33 | -0.33 | -0.32 | -0.31 | -0.31 | -0.31 | -0.30 | -0.29 | -0.26 | -0.26 | -0.25 | -0.21 | 0.00 |

Table A10: P-value matrix for TimeHUT and competing methods.

| | TimeHUT | AutoTCL | SelfDis | InfoTS$_s$ | FEAT | SMDE | InfoTS | TS2Vec | TNC | TS-TCC | T-Loss | TST | MCL | TF-C |
|---|---|---|---|---|---|---|---|---|---|---|---|---|---|---|
| **TimeHUT** | - | 0.0188 | 0.1073 | 0.0091 | <1e-4 | 0.001 | 0.0002 | <1e-4 | <1e-4 | <1e-4 | <1e-4 | <1e-4 | <1e-4 | <1e-4 |
| **AutoTCL** | 0.0188 | - | 0.8288 | 0.4123 | 0.0587 | 0.088 | 0.0362 | 0.0036 | <1e-4 | 0.0003 | 0.0004 | <1e-4 | <1e-4 | <1e-4 |
| **SelfDis** | 0.1073 | 0.8288 | - | 0.5769 | 0.0798 | 0.0382 | 0.0497 | 0.0019 | <1e-4 | 0.0002 | 0.0007 | <1e-4 | <1e-4 | <1e-4 |
| **InfoTS$_s$** | 0.0091 | 0.4123 | 0.5769 | - | 0.3805 | 0.432 | 0.0296 | 0.0191 | <1e-4 | <1e-4 | 0.0004 | <1e-4 | <1e-4 | <1e-4 |
| **FEAT** | <1e-4 | 0.0587 | 0.0798 | 0.3805 | - | 0.9808 | 0.3707 | 0.0174 | 0.0001 | 0.0002 | 0.0091 | 0.0001 | <1e-4 | <1e-4 |
| **SMDE** | 0.001 | 0.088 | 0.0382 | 0.432 | 0.9808 | - | 0.3869 | 0.1683 | <1e-4 | 0.0005 | 0.013 | <1e-4 | <1e-4 | <1e-4 |
| **InfoTS** | 0.0002 | 0.0362 | 0.0497 | 0.0296 | 0.3599 | 0.3869 | - | 0.2355 | 0.0003 | 0.0035 | 0.0057 | 0.0002 | <1e-4 | <1e-4 |
| **TS2Vec** | <1e-4 | 0.0036 | 0.0019 | 0.0191 | 0.0174 | 0.1683 | 0.2355 | - | 0.0026 | 0.0364 | 0.0898 | 0.0009 | <1e-4 | <1e-4 |
| **TNC** | <1e-4 | <1e-4 | <1e-4 | <1e-4 | 0.0001 | <1e-4 | 0.0003 | 0.0026 | - | 0.7051 | 0.4592 | 0.0207 | 0.0051 | <1e-4 |
| **TS-TCC** | <1e-4 | 0.0003 | 0.0002 | <1e-4 | 0.0002 | 0.0005 | 0.0035 | 0.0345 | 0.7051 | - | 0.6004 | 0.0201 | 0.0026 | <1e-4 |
| **T-Loss** | <1e-4 | 0.0004 | 0.0007 | 0.0004 | 0.0091 | 0.013 | 0.0057 | 0.0898 | 0.4592 | 0.6004 | - | 0.0047 | 0.0113 | 0.0001 |
| **TST** | <1e-4 | <1e-4 | <1e-4 | 0.0001 | <1e-4 | <1e-4 | 0.0002 | 0.0009 | 0.0207 | 0.0201 | 0.0047 | - | 0.2894 | 0.0005 |
| **MCL** | <1e-4 | <1e-4 | <1e-4 | <1e-4 | <1e-4 | <1e-4 | <1e-4 | 0.0051 | 0.0026 | 0.0113 | 0.2801 | 0.2894 | - | 0.0009 |
| **TF-C** | <1e-4 | <1e-4 | <1e-4 | <1e-4 | <1e-4 | <1e-4 | <1e-4 | <1e-4 | <1e-4 | <1e-4 | 0.0001 | 0.0005 | 0.0009 | - |

