# OpenReview forum: "Learning Time-Series Representations by Hierarchical Uniformity-Tolerance Latent Balancing"
_TMLR — Accepted by TMLR_

### Review · Reviewer_4fMT · 2025-08-03

**Summary Of Contributions:**

This work introduces a method to embed time series through a contrastive self-supervised learning methods. Through the design of a dedicated loss, the authors try to balance several behaviors to obtained good representations of the time series. Namely, they try to balance uniformity (maximization of the spreading of the representation) with tolerance (such that small variations of a sample have a close representation), while also maintaining a sufficient distance between negative pairs. They benchmark their method on several datasets and tasks, and compare it with several prior methods. Moreover, the authors add an ablation of the different part of the proposed loss.

**Audience:**

Yes

**Audience Explanation:**

This paper proposes a method to perform self-supervised learning on time series which is a lot of interests to a wide audience. The authors introduce a loss demonstrating state of the art results, and show that each component of their loss provides improvements through ablation studies, and do an extensive comparison with previous works. Thus, their results are also of interest to TMLR's audience.

**Claims And Evidence:**

Yes

**Claims Explanation:**

This work introduces a loss which allows to find embeddings of time series which balance several appealing properties, such as uniformity and tolerance. The paper is well written, the concepts are well explained, and the proposed loss is well motivated. The relation with previous works is also well described. The experiments demonstrate very good performance on classification tasks compared to previous works, and on several datasets. Moreover, these good results are also shown on anomaly detection tasks, where the results are competitive with previous works. Finally, there is also an extensive ablation showing the influence of each of the key components of the method, the analysis of the trade-off between uniformity and tolerance, and the sensitivity to different hyperparameters.

I am not very familiar with this literature, so I do not see major weaknesses. I think it is a nice work, using a well motivated loss to solve the task of self-supervised learning for time series, and which evaluated the method thoroughly, with comparisons with other methods on several benchmarks, and ablation studies. Maybe a slight weakness would be that it relies a lot on previous works, combining several components of different works. Nonetheless, it is done in a smart way and it is well justified.

**Requested Changes:**

I don't have major requested changes. A minor one would be to use with better care the \citep vs \citet in the citation format.


Typos:
- p2: "Following we summarize key recent papers in this area."

---

> ### Author Response · Authors · 2025-09-12
> **Response to Reviewer 4fMT**
>
> > Novelty
>
> We have now revised our introduction and contributions (Section 1, page 2) and method sections (Section 3.2, page 4) to explicitly mention our contributions:
>
> Our specific contributions are as follows: (1) Novel adaptation of temperature scheduling to hierarchical time-series contrasting (Equations 3-5): While TS2Vec (Yue et al., 2022) uses fixed temperature, we introduce periodic scheduling (Kukleva et al., 2023) within the hierarchical framework that balances uniformity-tolerance trade-offs dynamically during training. This addresses the limitation of contrastive loss, where fixed temperature parameters cannot adapt to the evolving representation space during training. (2) Adaptation of angular margins to temporal hierarchical contrasting (Equations 6-8):  Although angular margin is established in face recognition (Zhang et al., 2022a), we adapt angular margins for both temporal and instance-wise contrast losses, enforcing geometric margins between temporal segments within sequences, unexplored in prior time-series work. (3) Synergistic integration exceeding sum of parts: Our ablation (Table 5) shows the combination achieves 86.4% accuracy vs. 83.0% with only hierarchical loss, demonstrating improvements beyond the sum of parts. In summary, our method builds upon the hierarchical contrasting framework from TS2Vec (Yue et al., 2022), incorporating temperature scheduling inspired by Kukleva et al. (2023) and angular margin losses adapted from face recognition literature (Boutros et al., 2022; Zhang et al., 2022a). Our contribution lies in the novel adaptation and integration of these techniques for time-series representation learning to address the uniformity-tolerance problem.
>
> > Citation style
>
> We have carefully reviewed and corrected all citations throughout the manuscript:
> - Using \citep{} for parenthetical citations: "Recent work shows improvements (Yue et al., 2022)"
> - Using \citet{} for in-text citations: "Yue et al. (2022) demonstrate that..."
> These corrections appear on pages 3, 8, and throughout the manuscript.
>
> > Typo
>
> We have corrected this typo on page 2. The sentence now reads: "In the following, we summarize key recent papers in this area."

---

### Review · Reviewer_hKUk · 2025-08-04

**Summary Of Contributions:**

The paper proposes **TimeHUT**, a self-supervised framework for time-series representation learning that:

1. **Stacks three existing ingredients**
    - multi-scale (instance + temporal) contrastive objectives from **TS2Vec**;
    - a **periodic cosine temperature scheduler** that alternates between uniformity- and tolerance-favouring regimes;
    - an **angular-margin loss** (ArcFace-style) applied hierarchically to enforce a minimum angle between negatives.
2. **Reports experimental results** on 128 UCR and 30 UEA classification datasets and on Yahoo/KPI anomaly-detection benchmarks, showing small (≈1–2 pp) average gains over recent SSL baselines.
3. **Provides ablations** indicating that removing either the scheduler or the angular margin reduces classification accuracy.

**Key strengths**

- Well-executed engineering recipe that can be dropped into any TS2Vec-like codebase.
- Consistently non-negative performance deltas on a large public benchmark suite.

**Key weaknesses**

- ★ **Limited methodological novelty** – all three components are direct borrowings from prior work; TimeHUT contributes only their combination.
- **Marginal empirical improvements** relative to the added compute/hyper-parameter burden.
- **No analysis of efficiency** (training time, memory, etc.).

**Audience:**

Yes

**Audience Explanation:**

Researchers who care about pragmatic ways to squeeze a bit more accuracy out of TS2Vec-style pipelines may appreciate the plug-and-play recipe and the extensive benchmark sweep. The work is less attractive to readers looking for conceptual advances.

**Broader Impact Concerns:**

I do not see immediate ethical red flags.

**Claims And Evidence:**

No

**Claims Explanation:**

Novelty claims are not supported: the hierarchical contrast is directly adopted from TS2Vec; temperature scheduling and angular margins are well-established techniques; and there is no comparison to prior scheduler- or margin-based self-supervised learning methods applied to the same datasets.

**Requested Changes:**

1. **Clarify and justify novelty.** Rewrite the abstract and Related-Work sections to make explicit that the hierarchical contrast, temperature schedule, and angular-margin loss are all taken from prior work
2. **Provide an efficiency analysis.** Report wall-clock training time, peak GPU memory, and FLOPs/epoch for TimeHUT versus other baselines.

---

> ### Author Response · Authors · 2025-09-12
> **Response to Reviewer hKUk**
>
> We thank the reviewer for their thorough and constructive feedback. We address each concern below and apply the necessary changes to the revised manuscript, in which the revisions are marked in "blue":
>
> > Novelty
>
> We have now revised our introduction and contributions (Section 1, page 2) and method sections (Section 3.2, page 4) to explicitly mention the reference papers of each component:
>
> Our specific contributions are as follows: (1) Novel adaptation of temperature scheduling to hierarchical time-series contrasting (Equations 3-5): While TS2Vec (Yue et al., 2022) uses fixed temperature, we introduce periodic scheduling (Kukleva et al., 2023) within the hierarchical framework that balances uniformity-tolerance trade-offs dynamically during training. This addresses the limitation of contrastive loss, where fixed temperature parameters cannot adapt to the evolving representation space during training. (2) Adaptation of angular margins to temporal hierarchical contrasting (Equations 6-8):  Although angular margin is established in face recognition (Zhang et al., 2022a), we adapt angular margins for both temporal and instance-wise contrast losses, enforcing geometric margins between temporal segments within sequences, unexplored in prior time-series work. (3) Synergistic integration exceeding sum of parts: Our ablation (Table 5) shows the combination achieves 86.4% accuracy vs. 83.0% with only hierarchical loss, demonstrating improvements beyond the sum of parts. In summary, our method builds upon the hierarchical contrasting framework from TS2Vec (Yue et al., 2022), incorporating temperature scheduling inspired by Kukleva et al. (2023) and angular margin losses adapted from face recognition literature (Boutros et al., 2022; Zhang et al., 2022a). Our contribution lies in the novel adaptation and integration of these techniques for time-series representation learning to address the uniformity-tolerance problem.
>
> > Computational cost and efficiency
>
> We have now included a comprehensive efficiency analysis in Table 8, Section 5 (pages 12) on UCR Chinatown using NVIDIA RTX 3090:
>
> |Model   |Accuracy      |F1_score|Peak_gpu_memory (MB)| Mflops/epoch |Training time (s)|
> |--------|--------------|--------|------------------|-------------|--------------|
> |TFC	 |0.8663	    |0.8514 |1395.4             |   72.4       |	10.15    |
> |MF_CLR  |0.8861	    |0.8726 |1376.6	            |   52.3       |	9.55     |
> |CPC     |0.9301        |0.9181 |1377.4             | 	120.2      |	14.2     |
> |TLoss	 |0.9511        |0.8191 |1594.4             | 	56.3       |	164.3    |
> |TS2vec	 |0.9651        |0.9684 |1330.6             | 	51.2       |	9.36     |
> |CoST    |0.9704        |0.9609 |1420.6             | 	240.6      |	19.58    |
> |TNC     |0.9772        |0.9472 |1414.1	            |   192.1	   |    35.91    |
> |TimesURL|0.9744	    |0.9756 |1392.7             |	113.4      |	22.62    |
> |TS_TCC  |0.9831        |0.9784 |1399.4             | 	70.4       |   	13.14    |
> |TimeHUT |0.9854        |0.9826 |1398.4             |	69.1       |    12.95    |
>
> TimeHUT's training time (12.95s) compared to TS2Vec (9.36s) is justified by accuracy and F1-score improvements.
> On this particular dataset, TS-TCC shows comparable results to TimeHUT; however, according to Table 1, TS-TCC's average accuracies on 125 UCR and 29 UEA are (75.7, 68.2), which are lower than TimeHUT (86.4, 77.3).
>
> > Comparison with other schedulers
>
> We have also compared different schedulers using TimeHUT, showing that our Hierarchical-Scheduler achieves the best accuracy-efficiency trade-off. The base hyperparameters for all schedulers are min_tau: 0.1, max_tau: 0.75, and t_max: 10. Table 7 (Section 5, page 12) shows the experiments for the Chinatown and AtrialFibrillation datasets, reporting accuracy, training time, and peak GPU memory values.
> |                                      |       Chinatown        |        AtrialFibrillation
> | Scheduler              | HyperParams | Acc  | Time(s)| GPU(MB)|     Acc | Time(s)| GPU(MB) |
> |------------------------|-------------|------|--------|--------|---------|--------|---------|
> | Exponential            | Decay=0.95  | 0.980| 12.83  | 1376.8 |   0.417 | 20.32  | 3081.4  |
> | Sigmoid                | Steep=1     | 0.979| 14.11  | 1421.5 |   0.412 | 20.31  | 3901.2  |
> | Warmup-Cosine          | Warmup=2    | 0.962| 12.43  | 1414.9 |   0.407 | 19.76  | 2755.3  |
> | Sawtooth-Cyclic        | Cycle=T/3   | 0.979| 12.48  | 1392.1 |   0.413 | 19.93  | 3551.6  |
> | Logarithmic            | Offset=1    | 0.980| 12.85  | 1366.2 |   0.425 | 20.18  | 3405.1  |
> | Step-Decay             | Gamma=0.5   | 0.977| 12.41  | 1406.3 |   0.468 | 19.89  | 3385.9  |
> | Cosine-Restarts        | Period=5    | 0.974| 12.58  | 1387.7 |   0.446 | 20.57  | 3249.2  |
> | Hyperbolic-Tangent     | Steep=2     | 0.973| 12.43  | 1393.8 |   0.467 | 20.72  | 3251.1  |
> | Hierarchical-Scheduler | --          | 0.985| 12.95  | 1398.4 |   0.534 | 20.13  | 3171.7  |

---

### Review · Reviewer_K7oc · 2025-08-19

**Summary Of Contributions:**

The authors propose a method called TimeHUT for learning time-series representations by hierarchical uniformly-tolerance balancing of contrastive representations. Two distinct losses are proposed to learn strong representations for balancing between uniformly and tolerance in the embedding space. The authors conducted experiments on 128 UCR and 30 UAE datasets for univariate and multivariate classification, as well as Yahoo and KPI datasets for anomaly detection, which shows the effectiveness of TimeHUT.

**Audience:**

Yes

**Audience Explanation:**

This manuscript deals with balancing between uniformality and tolerance in time series contrastive learning which is now a hot topic im machine learning.

**Broader Impact Concerns:**

I did not find any obvious ethical and societal issues.

**Claims And Evidence:**

Yes

**Claims Explanation:**

The modeling capability of the proposed method TimeHUT is well supported by the empirical performance.

**Requested Changes:**

1. I find that Figure 1 is not easy to read. I suggest the authors polish the caption and use more precise components in the figure.

2. Although TMLR does not require each paper to be highly novel, this work's technological novelty can be made clearer and deeper by first providing a clear discussion of modeling insights from related works and then discussing some theoretical aspects of TimeHUT.

3. The authors provide a block of PyTorch-like pseudo-code for TimeHUT. However, I find it still inconvenient for readers to conduct sufficiently reproducible experiments for many reasons, such as detailed parameter settings. Why not directly share the code on platforms like GitHub?

4. The use of hierarchical information may add significant computational complexity. I suggest that the authors discuss the time cost of TimeHUT.

5. Are there failure cases where TimeHUT significantly adds a computational burden or introduces inappropriate inductive biases that hinder efficiency or effectiveness?

---

> ### Author Response · Authors · 2025-09-12
> **Response to Reviewer K7oc**
>
> We sincerely thank the reviewer for their constructive feedback and positive assessment of our work. We address each concern below and apply the necessary changes to the revised manuscript, in which the revisions are marked in "blue":
>
> > Fig. 1 caption
>
> We have revised Figure 1 (Section 3, page 4) with the following enhancements: (1) Added detailed component labels (e.g., "Hierarchical Contrastive", "Temperature Scheduler, and Hierarchical Angular Margin"). (2) Improved visual clarity with color coding and clearer connections. (3) Expanded caption with explicit description of each component as follows: "Figure 1: Overview of TimeHUT architecture. (a) Input time-series $X$ (with samples $i$, $j$, and $k$ shown in different colors) are randomly cropped into two overlapping subseries. (b) The encoder processes these subseries, followed by hierarchical contrasting at both temporal and instance-wise levels through max pooling, computing both temporal contrast loss (within-sample across time) and instance-wise contrast loss (across samples). (c) The temperature scheduler dynamically balances uniformity-tolerance trade-offs ($L_{HierSch}$), while angular margin loss enforces geometric separation between positive and negative pairs ($L_{HierAng}$). The final loss combines both components."
>
> > Clarification of novelty
>
> We clarify the modeling insights and theoretical contributions (now in Section 3.2, pages 4):
>
> Our method builds upon the hierarchical contrasting framework from TS2Vec (Yue et al., 2022), incorporating temperature scheduling inspired by Kukleva et al. (2023) and angular margin losses adapted from face recognition literature (Boutros et al., 2022; Zhang et al., 2022a).
> Our contribution lies in the novel adaptation and integration of these techniques for time-series representation learning to address the uniformity-tolerance problem. While TS2Vec uses fixed temperature in hierarchical contrasting, we introduce periodic scheduling (Kukleva et al., 2023)  within the hierarchical framework that balances uniformity-tolerance trade-offs dynamically during training. This addresses the limitation in contrastive learning where fixed temperature parameters cannot adapt to the evolving representation space during training. The temperature parameter $\tau$ controls the boundaries between negative and positive pairs based on theoretical studies in  Kukleva et al. (2023) and Wang & Isola (2020). When $\tau$ is small, the gradients are sharper and promote uniformity by maximizing average distances to the nearest neighbors. When $\tau$ is large, the gradients become smoother and promote tolerance by allowing tighter clustering. TimeHUT additionally employs instance-wise and temporal hierarchical contrastive angular margin losses to enforce coherence among proximate segments, while distinctly separating them from non-neighboring ones within the same time-series sample. This adaptation of angular margins for both temporal and instance-wise contrast losses is unexplored in prior time-series work. The combination of hierarchical temperature scheduling with the angular margins allows us to navigate the uniformity-tolerance trade-off more effectively than fixed-temperature approaches.
>
> > Code instead of pseudo-code
>
> We have released our complete code implementation at https://anonymous.4open.science/r/THmodelsubmission-A814/README.md
> (Section 1, page 2, end of contributions list).
>
> > Computational complexity
>
> We have provided a detailed computational analysis in Section 5, page 11, Table 6, using the UCR Chinatown dataset (NVIDIA RTX 3090, Batch_size=8, Epochs=200):
>
>
> | Scenario                   | Accuracy	| Training_Time(s) | Peak_GPU_Memory (MB) | MFLOPs/Epoch |
> |----------------------------|----------|------------------|----------------------|--------------|
> |Baseline                    | 0.9651	|   9.36           |     1330.6           |   51.2   	 |
> | +HierAng_Instance          | 0.9823	|   11.09          |     1315.9	          |   60.5	     |
> | +HierAng_Temporal          | 0.9795	|   12.34	       |     1341.6	          |   62.7       |
> | +HierAng_Both	             | 0.9828	|   12.49	       |     1367.6	          |   65.3       |
> | +HierScheduler             | 0.9815	|   9.53	       |     1332.2	          |   58.5	     |
> | +HierAng_Both+HierScheduler| 0.9854	|   12.95          |     1398.4	          |   69.1	     |
>
> The training time overhead (12.95s vs 9.36s) yields +2.03% accuracy gain, demonstrating a favorable cost-benefit trade-off. TimeHUT adds minimal overhead while providing accuracy gains on 128 UCR datasets (86.4% vs 83.0% without our components, Table 5).

---

> ### Author Response · Authors · 2025-09-12
> **Response to Reviewer K7oc**
>
> > Failure cases in terms of efficiency or effectiveness
>
> We have added a discussion of failure cases in Section 5, pages 13-14.
>
> When the time-series length is very short (T<20), the hierarchical contrasting has insufficient temporal context, resulting in suboptimal performance. In addition, for very long sequences (T>10000), the quadratic complexity with respect to sequence length becomes prohibitive. On the other hand, angular margins may be too restrictive when few samples exist per class. In our experiments, TimeHUT showed reduced improvements on datasets like "ScreenType" (accuracy: 0.403) and "PhonemeSpectra" (accuracy: 0.243), which have either very short sequences or complex multivariate distributions where angular margins may overseparate natural clusters.
>
> > Comparison with other schedulers
>
> We have also compared different schedulers using TimeHUT, showing that our Hierarchical-Scheduler achieves the best accuracy-efficiency trade-off. The base hyperparameters for all schedulers are min_tau: 0.1, max_tau: 0.75, and t_max: 10. Table 7 (Section 5, page 12) shows the experiments for the Chinatown and AtrialFibrillation datasets, reporting accuracy, training time, and peak GPU memory values.
> |                                      |       Chinatown        |        AtrialFibrillation
> | Scheduler              | HyperParams | Acc  | Time(s)| GPU(MB)|     Acc | Time(s)| GPU(MB) |
> |------------------------|-------------|------|--------|--------|---------|--------|---------|
> | Exponential            | Decay=0.95  | 0.980| 12.83  | 1376.8 |   0.417 | 20.32  | 3081.4  |
> | Sigmoid                | Steep=1     | 0.979| 14.11  | 1421.5 |   0.412 | 20.31  | 3901.2  |
> | Warmup-Cosine          | Warmup=2    | 0.962| 12.43  | 1414.9 |   0.407 | 19.76  | 2755.3  |
> | Sawtooth-Cyclic        | Cycle=T/3   | 0.979| 12.48  | 1392.1 |   0.413 | 19.93  | 3551.6  |
> | Logarithmic            | Offset=1    | 0.980| 12.85  | 1366.2 |   0.425 | 20.18  | 3405.1  |
> | Step-Decay             | Gamma=0.5   | 0.977| 12.41  | 1406.3 |   0.468 | 19.89  | 3385.9  |
> | Cosine-Restarts        | Period=5    | 0.974| 12.58  | 1387.7 |   0.446 | 20.57  | 3249.2  |
> | Hyperbolic-Tangent     | Steep=2     | 0.973| 12.43  | 1393.8 |   0.467 | 20.72  | 3251.1  |
> | Hierarchical-Scheduler | --          | 0.985| 12.95  | 1398.4 |   0.534 | 20.13  | 3171.7  |

---

> > ### Comment · Reviewer_K7oc · 2025-09-16
> >
> > Many thanks for the detailed response which addresses my concerns.

---

### Decision · Action_Editor_F5qh · 2025-09-21

**Recommendation:** Accept as is

**Audience:**

Yes

**Audience Explanation:**

This paper is relevant for learning representation of time series with self-supervision, which is relevant both as a standalone method, but also for applications.

**Claims And Evidence:**

Yes

**Claims Explanation:**

This paper proposes a method to learn time series representations using contrastive representations. The approach is thoroughly evaluated and the authors have added experiments as requested by reviewers during the rebuttal. These should be included in the final revision. There was some confusion around Figure 1. I think it's ok, but I agree that clarity could be improved, so I would encourage the authors to try. The novelty of the method was also questioned (a combination of known things), but the performance improvement was deemed interesting.